



# A retrieval of $xCO2$ from ground-based mid-infrared NDACC solar absorption spectra and comparison to TCCON

Rafaella Chiarella[1], Matthias Buschmann[1], Joshua Laughner[2], Isamu Morino[3], Justus Notholt[1], Christof Petri[1], Geoffrey Toon[2], Voltaire A Velazco[4], and Thorsten Warneke[1]

[1]Institute of Environmental Physics, University of Bremen, Bremen, Germany
[2]Jet Propulsion Laboratory, California Institute of Technology, Pasadena, CA, USA
[3]Earth system Division, National Institute for Environmental Studies (NIES), Onogawa 16-2, Tsukuba, Ibaraki 305-8506, Japan
[4]School of Earth Atmospheric and Life Sciences, University of Wollongong, NSW 2522, Australia. *Now at: Deutscher Wetterdienst (DWD), Meteorological Observatory Hohenpeissenberg, 82383, Germany

**Correspondence:** Rafaella Chiarella (rachi@iup.physik.uni-bremen.de)

**Abstract.**

Two global networks of ground-based Fourier transform spectrometers are measuring abundances of atmospheric trace gases that absorb in the near and mid infrared, Network for the Detection of Atmospheric Composition Change (NDACC) and Total Carbon Column Observing Network (TCCON). The first lacks a $CO2$ product, therefore this study focuses on developing a $xCO2$ retrieval method for NDACC from a spectral window in the 4800 $cm^{-1}$ region. This retrieval will allow to extend ground-based measurements back in time, which we will demonstrate with historical data available from Ny-Ålesund. This region is covered by both TCCON and NDACC, which is an advantage for collocated comparisons where available. The results are compared with collocated TCCON measurements of column-averaged dry-air mole fractions of $CO2$ (denoted by $xCO2$) in Ny-Ålesund, Svalbard and only TCCON in Burgos, Philippines. We found that it is possible to retrieve $xCO2$ from NDACC spectra with a precision from $0.2\%$ . The comparison between the new retrieval to TCCON showed that the sensitivity of the new retrieval is high in the troposphere and lower in the upper stratosphere, similar to TCCON, and that the seasonality is well captured. We determined an optimal retrieval setup covered in section 7.

## 1 Introduction

Carbon Dioxide ($CO2$) is the most important anthropogenic greenhouse gas. Fossil fuel combustion and deforestation are the main net sources, the ocean and terrestrial ecosystems currently act as net sinks, absorbing approximately half of the anthropogenic emissions (IPCC, 2022). Remote sensing measurements provide a column integral, that is less affected by vertical transport and local sources. The $CO2$ column derived from ground based Fourier transform infrared (FTIR) spectrometry from near infrared radiation (NIR) solar absorption spectra can achieve a precision better than 0.25% (Wunch et al., 2011a).

The Total Carbon Column Observing Network (TCCON), a worldwide network, has currently 26 operational measurement sites since 2004. The network was founded for satellite validation validation and remotely measures abundances ofCO2, CO, CH4, O2 and other molecules absorbing in the near-infrared covering the spectral range from 4000 $cm^{-1}$ to 11000 $cm^{-1}$.





The Network for the Detection of Atmospheric Composition Change Infrared Working Group (NDACC-IRWG) was founded in 1991 and has a total of 25 sites. The network provides total column abundances and profiles of several atmospheric constituents, with a focus on O3 cycle (De Maziere et al., 2018) retrieved from the mid infrared covering the spectral region 2000 to 5000 $cm^{-1}$ uses optical filter to limit the spectral range per recorded sppectru,. The NDACC official filters are listed in the Appendix of Blumenstock et al. (2021).

NDACC trace gases, in contrast to TCCON, don't include CO2. An extension of the retrieval capabilities of NDACC spectra to include a CO2 product would expand the total column products in time and also expand the spatial coverage. In this study one set of historical data, for Ny-Ålesund is presented, where the CO2 spectra date back to 1997. Two major challenges have to be faced on the way to an NDACC xCO2 product. TCCON retrieval method uses the ratio of CO2 from the 6300 $cm^{-1}$ band and O2 from the 7885 $cm^{-1}$ band (**?**), in contrast there are no O2 absorption lines present in the NDACC MIR spectra; therefore there is no proxy for the dry-air column to which to take a ratio. N2 lines are present in the 4800 $cm^{-1}$ region, however, the lines are too weak. Additionally the presence of multiple interfering gases hinders the use of broad spectral windows (Buschmann et al., 2016).

Previous efforts towards a xCO2 product using MIR spectra include Barthlott et al. (2015) and Buschmann et al. (2016). The xCO2 retrieval proposed by Barthlott et al. (2015) using four microwindows in the 2620 $cm^{-1}$ region was appropriate for long-term monitoring of instrument stability and consistency of trends of tropospheric species. However, due to the low sensitivity of the MIR averaging kernels to the surface, the proposal by Barthlott et al. (2015) was not applicable for shorter timescales therefore it did not provide new information on the carbon cycle, as the annual trends are already well understood. Similarly, Buschmann et al. (2016) presented an approach for the retrieval of xCO2 from several NDACC MIR micro windows in the 2620 to 3350 $cm^{-1}$ region. Nonetheless, this approach showed to have a strong sensitivity to the chosen a priori and that the tropospheric signal is damped in comparison to TCCON.

In this study a retrieval of the column average dry-air mole fraction of CO2 from MIR solar spectra in the 4800 $cm^{-1}$ region is described. We will discuss the different retrievals, their sensitivities and dependence on the a priori information and airmass factor. We will present the averaging kernels, and their influence on the retrieval and our approach to an error budget, covering standard errors, the diurnal variation and sensitivity to several error sources and finally their application to Ny-Ålesund historical data.

In the next section information on the selected sites is given and the window and the retrieval used are described. Then in the third to fifth sections the influence of the averaging kernels, the a priori and airmass factor on the retrieval is investigated and compared to TCCON, respectively. The sixth section includes the error budget and error source tests. In the seventh section the time series for both Burgos and Ny-Ålesund historical data is presented. Finally, in section nine an acquisition strategy is proposed, followed by a conclusion about the retrieval method.



## 2 Measurements and Methods

### 2.1 Location and Measurements

Data from two sites were used, Ny-Ålesund in the Arctic and Burgos in the tropics. Ny-Ålesund provides the collocated measurements of NDACC and TCCON. On the other hand Burgos gives us the opportunity to assess the retrieval in an atmosphere with higher temperature and water content.

The measurements in Ny-Ålesund (Spitzbergen, 78.92°N, 11.92°E) were performed using the Bruker IFS 120-5 HR instrument with Indium Gallium Arsenide (InGaAs) and liquid nitrogen cooled Indium Antimonide (InSb) detectors. In NDACC spectra are recorded with the InSb detector since 1992 and cover the mid-infrared spectral region with an optical path difference (OPD) of 180 cm, giving a spectral resolution of approximately $0.005 \ \mathrm{cm^{-1}}$. The acquisition is performed with different band-pass filters. The filter of interest for this study transmits from $4044 \ \mathrm{cm^{-1}}$ to $4822 \ \mathrm{cm^{-1}}$, will be referred to as filter 4433 as its center wave number, and started being used in 1996. On the other hand, TCCON uses the InGaAs detector for its measurements since 2004 covering the spectral region $4000 \ \mathrm{cm^{-1}}$ to $11000 \ \mathrm{cm^{-1}}$ (Wunch et al., 2011a).

Burgos (Philippines, 18.53°N 120.65°W) has a warm and humid climate. There, only TCCON InGaAs measurements are performed with a FTS model Bruker 125 HR. The operations started in 2017 (Morino et al., 2018; Velazco et al., 2017). The tropics face different challenges with a higher water vapour content along the light path and a higher tropopause.

The spectra available were separated in two groups, InSb (from NDACC) and InGaAs (from TCCON), both were used to retrieve xCO2 from the new window and compared to the xCO2 retrieved from the two TCCON CO2 windows from InGaAs spectra.

Additionally, collocated TCCON and aircraft profiles from campaigns over TCCON stations and AirCore were used where available to do a comparison with in situ data. This allows to determine the systematic biases in the spectroscopy of the column measurement as is similarly done for TCCON windows in the recent GGG2020 release (Laughner, J. L. et al., 2022).

### 2.2 CO$_2$ window in the MIR

The mid-infrared region not only has CO2 absorption lines but also other gases with strong absorption lines. The high resolution measurements of NDACC allow for the use of narrower windows or single lines. The selected spectral window is centred at $4790 \ \mathrm{cm^{-1}}$ and has a $20 \ \mathrm{cm^{-1}}$ width, hence we will refer to it as **w4790**. It contains water lines but minimal interference from other gases.

To choose the windows, we made use of information on the NDACC filters and the intensity of the transitions of CO2 in the region. The first step to select prospect windows was to study the NDACC spectra available from Ny-Ålesund and select which of the available regions will be used. The region of $4800 \ \mathrm{cm^{-1}}$ acquired using the filter 4433 was selected because it contains a strong band of CO2 corresponding to the transition $21113 \rightarrow 01101$ for the main isotope and weaker bands for the other isotopes (Toth et al., 2008). This is referred to as a "hot" band because it contains a transition between two excited vibrational states. The population of the initial state is dictated by the Boltzmann distribution, this makes the band temperature dependent, as the intensity is proportional to the population of the excited initial state Buckingham (1976). With a ground-state energy

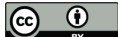

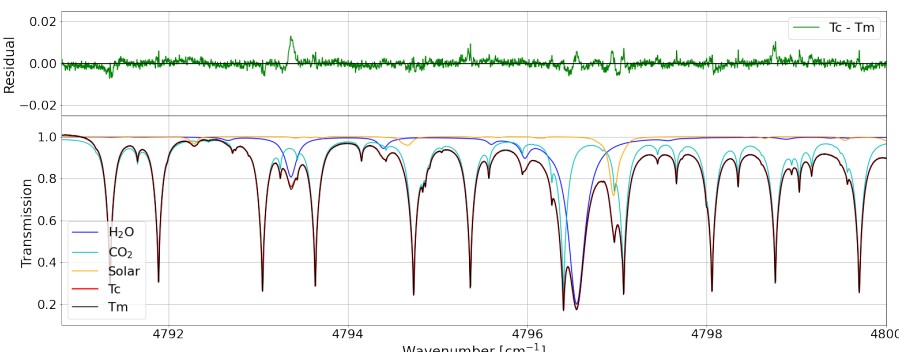

**Figure 1.** An example of the computed transmittance (Tc) and the measured transmitance (Tm) for the w4790 window, CO2 absorption lines and other gases, and the residual (Tm-Tc) for a typical InSb spectrum acquired in Ny-Ålesund.

$E'' = 858 \ \mathrm{cm}^{-1}$ the estimated theoretical error for a 2 K error in temperature is between 1.2 to 1.8% of the retrieved CO2. However from tests performed, the temperature sensitivity is approx. 1.5%/K (see subsection 5.2). The CO2 transitions and the intensity of each line was taken from the line parameters from the GFIT atm.161 line list, which is based on the HITRAN 2016 (high-resolution transmission molecular absorption) database (**?**). The line list and an evaluation of the temperature and
the impact of using isotopes for the retrieval are addressed in the appendix.

An example of the spectral fit is shown in fig. 1 where we can observe how the measured spectra fits the CO2 spectroscopic lines. On the left, there is an overlap between measured transmittance (TM) and CO2, however it slopes down towards the left where we can see a considerable amount. This is caused by the filter used to record this spectra. The w4790 window is located near the end of the filter 4433, where the transmission has started to decline (see Appendix C).

To provide the best fit to the measured spectrum and minimize the residuals, other gas profiles and other parameters are scaled and adjusted. The gas mole fraction is retrieved from these scaled profiles (Wunch et al., 2015). The parameters fitted are: continuum level, continuum tilt, continuum curvature, frequency shift and solar lines. The gases fitted are: CO2, H2O, HDO, CH4 and N2O. See in Appendix A the full window fitting parameters used in the retrieval.

For comparison the TCCON xCO2 data from both locations was used (Laughner, J. L. et al., 2022). These windows are
centred at 6220.00cm$^{-1}$ and 6339.50 cm$^{-1}$ with spectral widths of 80 cm$^{-1}$ and 85 cm$^{-1}$ respectively (Wunch et al., 2011a). Additionally, TCCON is introducing a new window in the 4800 cm$^{-1}$ region called lCO$_2$, for more details refer to the Appendix G. The full dataset from TCCON is available for download at https://tccondata.org/.

In this study we work with two types of spectra, that we will denote with the detector used to record, **InGaAs** and **InSb**. Following this, we have three windows used for retrievals, the w4790 and the two TCCON windows around the **6300** cm$^{-1}$.
Details of the three xCO2 retrievals are shown in the Table 1.



**Table 1.** The names of the 3 different xCO2 retrievals, with the spectra details, the resolution of each spectra used.

| Name | Details | Resolution $\text{cm}^{-1}$ |
|---|---|---|
| w4790 InSb | from NDACC spectra using w4790 | 0.005 |
| w4790 InGaAs | from TCCON spectra using w4790 | 0.02 |
| 6300 | from TCCON spectra using TCCON windows | 0.02 |

## 2.3 Retrieval Method

The retrieved xCO2 in this study is obtained by the profile scaling algorithm GFIT (version 5.28 in GGG2020) nonlinear least-squares fitting algorithm, which is also used for the TCCON xCO2 retrievals. A full description is available in Wunch et al. (2015)

The CO2 column averaged dry-air mole fraction retrieved by TCCON is calculated using O2 obtained from the band at 7885 $\text{cm}^{-1}$. The well mixed O2 is used to estimate the total dry-air column. The column averaged dry air mole fraction is derived by dividing the CO2 by the O2 (Wunch et al., 2011a). For the purpose of this retrieval the O2 mole fraction is assumed constant.

$$xCO_2 = \frac{VC_{CO_2}}{VC_{O_2}} \times 0.2095 \tag{1}$$

w4790 xCO2 was also retrieved using the GFIT software, for both InSb and InGaAs spectra. However, InSb spectra used
here don't contain any O2 window. For that reason the retrieval post processing for w4790 in GFIT doesn't include the air mass correction or the in situ correction, whereas 6300 and TCCON does include both. The mole fraction for the MIR spectra was calculated using the dry-air column abundance inferred from the surface pressure (Wunch et al., 2011a) with the following equation:

$$xCO_2 = \frac{VC_{CO_2}}{\frac{P_s N_A}{m_{air}^{dry} \{g\}} - \frac{VC_{H_2O} m_{H_2O}}{m_{air}^{dry}}} \tag{2}$$

$VC_{CO_2}$ is the CO2 vertical column from the GGG2020 output from w4790, $VC_{H_2O}$ is the vertical column from the GGG2020 output for water vapour from the window at 4576.85 $\text{cm}^{-1}$ of width 1.90 $\text{cm}^{-1}$, $P_s$ is the surface pressure in hPa, $Na = 6.0221415 \ 10^{23}$ molecules/mole, the Avogadro number; $m_{air}^{dry} = 28.9644$ g/mole, mass of dry air; $m_{H_2O} = 18.01534$ g/mole, the mass of water and the column-averaged gravitational acceleration $\{g\} = 9.81 \text{ms}^{-2}$.

These are two big important differences between the 6300 and the w4790 retrievals, the calculation of the mole fraction using O2 or pressure and the inclusion of an airmass and in situ corrections or lack of respectively.



The retrieved mole fraction is quality controlled following a similar flagging as TCCON (Wunch et al., 2011a). Data points with a high solar zenith angles (SZA) ($>83°$ as used by TCCON in Burgos) and/or high relative retrieval errors ($10\%$ of the average) were discarded; spectra with negative error output were also removed. Finally, data points with high relative retrieval error ($>20\%$ of the average) of H2O were filtered out.

For the time series, an error-weighted daily mean and the standard deviation of the daily mean was calculated. Because InSb spectra recorded each day on the filter covering w4790 is not abundant (usually less than 20), the weighted daily mean of the 6300 was used to calculate the standard deviation.

$$\bar{x}^{daily} = \frac{\sum_N^i \frac{x_i}{\epsilon_i^2}}{\sum_N^i \frac{1}{\epsilon_i^2}} \tag{3}$$

$$\sigma_{w4790} = \sqrt{\frac{\sum_N^i \left(\frac{x_i - \bar{x}^{daily}}{\epsilon_i}\right)^2}{\sum_N^i \frac{1}{\epsilon_i^2}}} \tag{4}$$

The error budget was approximated by using the standard deviations of the daily means (eq. 4) and the standard error to better reflect the uncertainty in the daily mean which is affected by the number of measurements. In addition, the diurnal variation:

$$DV = \left(\frac{x}{\bar{x}_{daily}} - 1\right) \times 100 \tag{5}$$

was calculated to determine the precision following **?**.

In the next sections several sensitivity tests and error budgets are presented to determine the quality of the retrieval. For these tests spectra from Burgos during 2017 and 2018 and Ny-Ålesund during 2016 to 2018 were used. Then the xCO2 time series from the w4790 window for Ny-Ålesund is shown.

## 3   Averaging Kernels

In this section the effect of the different averaging kernels on the retrieved xCO2 is evaluated, by comparing two different retrievals (Rodgers and Connor, 2003). For a perfect column measurement, the column averaging kernel should be 1.0 at all altitudes, but in practice, there is a greater sensitivity to some altitudes than others. The column averaging kernels depend on the retrieval method and the choice of parameters to be fitted and the solar zenith angle (Wunch et al., 2011a). The main difference in the averaging kernels originates in the shape and depth of the absorption lines. In this case the w4790 InSb CO2 averaging kernels show a good sensitivity in the troposphere and it decreases towards the upper atmosphere. Good sensitivity remains towards the stratosphere because the lines used are weak and don't saturate.





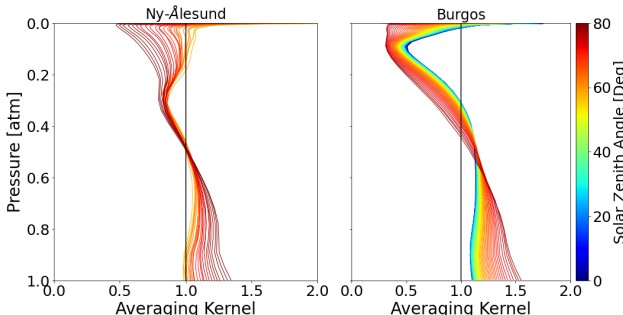

**Figure 2.** w4790 averaging kernels averaged to 1° for all InGaAs Ny-Ålesund on the left and Burgos on the right.

In figure 2 we see the averaging kernels for w4790 InGaAs in Ny-Ålesund, those corresponding to w4790 InSb spectra look

very similar (see supplemental Figure H1 in appendix).

We observe that the averaging kernels for Burgos look different to Ny-Ålesund, however, follow a similar curvature but at different altitude. When seeing the averaging kernels plotted against altitude (see in supplemental plots in Appendix) the curve towards less than one occurs around the 10 km for Ny-Ålesund around 15 km in Burgos, but the overall shape is similar. This is probably due to the difference in the height of the tropopause in both locations.

## 4   Sensitivity studies

### 4.1   A priori influence

As described by Rodgers (2000), the retrieved quantity $\hat{x}$ relates to the true atmospheric quantity $x$, the a priori $x_a$ and the averaging kernel $A$ by the following:

$$\hat{x} - x_a = A(x - x_a) + \epsilon_x \tag{6}$$

where $\epsilon_x$ is the random and systematic error term. The a priori profile will influence the retrieved xCO2 depending on how much the measured information can constrain the retrieval (Rodgers, 2000). The prior profiles of GGG2020 are produced using the Goddard Earth Observing System Forward Product for Instrument Teams (GEOS-5 FP-IT or GEOS FP-IT) reanalysis product that has a temporal resolution of three hours. The full description is found in Laughner et al. (2022).

The use of a priori profiles that are close to the true atmosphere can give the impression of a successful retrieval, when the

information content of the measured quantity is not sufficient. To discard this possibility, the influence of the a priori needs to be evaluated with a set of test retrievals performed with a modified a priori. In this section we compare the xCO2 retrieved using the modified a priori, which is initially fixed to 400 ppm at all altitudes and includes a linear increase of 0.1 ppm for each modelled profile (see Fig. H5 in the Appendix), with the xCO2 retrieved using the standard a priori also used in the time series and other tests. Apart from the different a priori, the same retrieval and post processing parameters have been used as in

the original retrievals with the standard GFIT a priori.





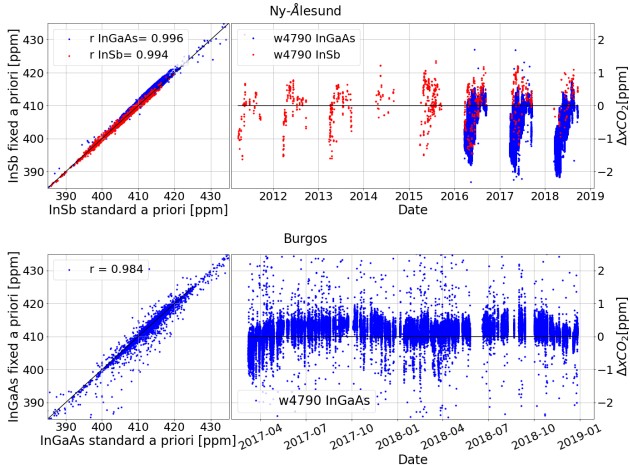

**Figure 3.** On the top Ny-Ålesund correlations of w4790 InGaAs xCO2, in blue, and w4790 InSb xCO2, in red on the left and the difference between them on the right. On the bottom Burgos correlation of w4790 InGaAs xCO2 with standard and fixed a priori on the left and the difference between them on the right

For both locations the correlation coefficient $r$ is above 0.95 which means a strong correlation. From figure 3 we observe the scattering for Burgos is higher than for Ny-Ålesund, likely due to the higher humidity and temperature given that the window is temperature sensitive. For Ny-Ålesund we see in the latest years a tendency to shift towards lower $\delta$ xCO2. The modified a priori used for these tests include a linear increase of xCO2. This increase might not correctly represent the atmospheric evolution and results in a over estimation. For both locations, the $\Delta$xCO2 is mostly smaller than 2 ppm.

## 4.2 Solar zenith angle dependence

The w4790 xCO2 has an airmass-dependent artifact and an airmass independent artifact similarly to the TCCON xCO2. The airmass-dependent artifacts are primarily caused by systematic errors introduced by spectroscopic inadequacies in the linelist and instrumental problems Wunch et al. (2011a).

From Wunch et al. (2011a), the TCCON xCO2 product has a SZA or airmass-dependent artefact causing the retrieval to be larger, by approximately 1% at 20° SZA than at 80° SZA . To correct this, TCCON derives and applies a single empirical airmass correction.

In this study we did not apply the airmass correction to the w4790 retrievals shown, to determine if the retrieval of xCO2 has to be corrected for airmass artefacts. We show in Figure 4, $xCO2 - \bar{x}CO2$ against SZA, where $\bar{x}CO2$ is the daily mean. Under the assumption that, on a given day, any variation of xCO2 that is symmetrical around noon is an artefact and anti-symmetrical variations are real, the dependence is observed (Wunch et al., 2011).





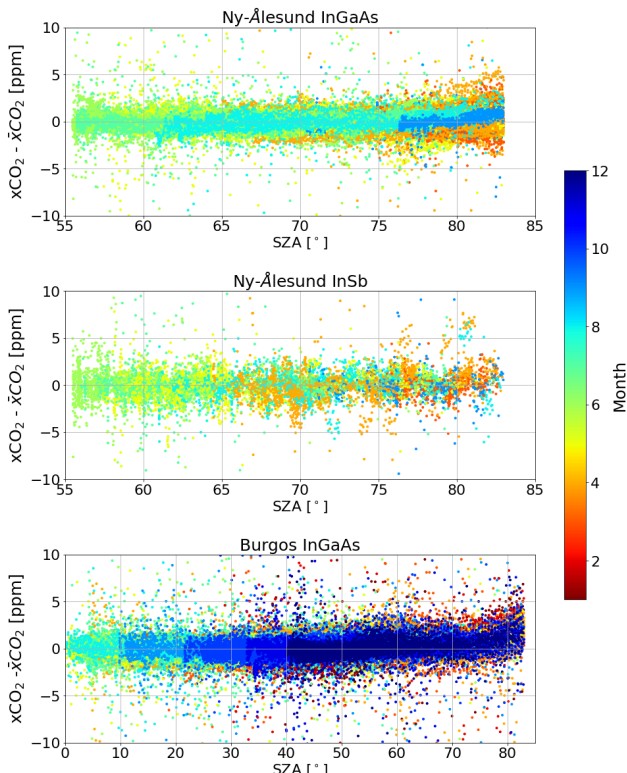

**Figure 4.** TopNy-Ålesund w4790 InGaAs xCO2 minus the daily mean. Middle: Ny-Ålesund w4790 InSb xCO2 minus the daily mean. Bottom: Burgos w4790 InGaAs xCO2 minus the daily mean.

For w4790 InGaAs, we can see that there is an increase of the scattering of xCO2 in SZA larger than 75° for Ny-Ålesund. Some values can be approximately 1% larger at 83° than at SZA lower than 75°, others remain within the range $\pm\ 2\ \mathrm{ppm}$. w4790 InSb seems to remain mostly unchanged for the SZA range (50° to 83°), but there is the possibility that this is due to the low number of data points.

For Burgos increase of all xCO2 values for SZA larger than 50°, with values around 83° being 1% larger than for SZA values below 50°. This asymmetry around zero indicates an airmass dependence for large SZA. On the other hand we found that for the 6300 window, prior to the airmass correction, the tendency is consistent with the findings of Wunch et al. (2011a) where at lower SZA the values are higher than at high SZA (see Fig. H6 and H7 in the appendix). This indicates that both retrievals have an air mass dependence, especially for high SZA, but the observed artefacts are different.

### 4.3 Site to site consistency

To investigate if there is a scaling between 6300 and w4790 xCO2, if it was consistent between both locations used and to emulate a correction via a scaling factor, a ratio between 6300 and w4790 for each data point was calculated.





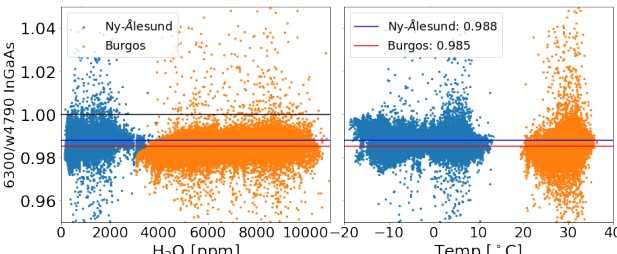

**Figure 5.** The ratios of Ny-Ålesund in blue and Burgos in orange plotted against H2O content on the left and against temperature on the right. The means of the ratio presented as lines, blue for Ny-Ålesund and red for Burgos.

The means of the ratios (shown in Fig. 5) were calculated with their corresponding standard deviation, these values are presented in table 2. The difference between the ratios' means are within the error, indicating a good consistency between both
sites. These values also serve to determine the scaling factor (SF) to be applied to both w4790 retrievals shown in section 7. To determine a global SF, all data points from Ny-Ålesund and Burgos were used to calculate the mean. Table 2 shows the values for the ratios and the SF.

**Table 2.** The ratio means and standard deviations of w4790 InGaAs/6300.

| Name | Mean | $\sigma$ |
|---|---|---|
| Ny Ålesund | 0.9880 | 0.0050 |
| Burgos | 0.9852 | 0.0060 |
| Global SF | 0.9863 | 0.0058 |

### 4.4 Comparison with Aircraft profiles

The airmass-independent correction factor (AICF) is determined with colocated in situ profiles. The procedure for this comparison is similar to the calibration of TCCON described in (Wunch et al., 2010). Xluft is used to diagnose instrument or retrieval errors and is calculated as the ratio of equation 1 and equation 2 which are the two ways to calculate the dry column for a given gas. A higher dependence will result in an increased xCO2 bias.

The correlation between w4790 xCO2 and in situ xCO2 shown in Fig. 6 right plot shows a good performance of w4790 in the
sites evaluated. The correlation coefficient $R^2$ is 0.9995.

The w4790 has a slightly higher bias and scatter to the in situ xCO2 than the TCCON windows. This means that if xluft deviates much from the nominal 0.999 value, there will be an increased bias in xCO2 (see Fig. 6 center plot). There is a weak correlation between the bias and the SZA (see Fig 6 right plot). The variation with SZA is smaller than the scattering, therefore the lack of airmass correction is a minor component. The higher scatter is to be expected as the w4790 has higher sensitivity
towards the surface and this might be driven by the surface variability of CO2, that aircrafts and balloons can't capture, and





the temperature sensitivity of the window. These features make the w4790 single data point uncertainties larger than the those of 6300.

The mean ratio is 1.01 for the 6300 windows and 1.005 for w4790 window suggesting that the line strengths in the 6300 windows are biased lower than in the w4790. These bias, however, is going to be removed by using the airmass correction

factor for TCCON and the global scaling factor (SF) for w4790.

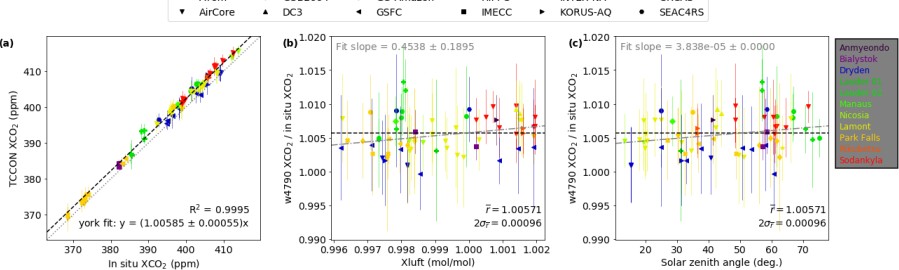

**Figure 6.** Left: Correlation between w4790 xCO2 and in situ xCO2. Centre: Ratio of w4790 xCO2 window / in situ xCO2 against xluft between 0.996 and 1.002. Right: Ratio of w4790 xCO2 window / in situ xCO2 against solar zenith angle.

## 5   Error Analysis

In this section we will look into the systematic and random errors of this retrieval. This analysis was performed on filtered data as described in section 2.3. A total of 3304 data points for InGaAs were flagged and removed, leaving 46307 usable points. For

InSb 121 data points were flagged and removed, leaving 1115 usable data points. In total, less than 10% of the original data points were filtered out.

As described by equation 4, an error weighted standard deviation of the daily mean, $\sigma_{day}$ was calculated for the years presented in the previous sections. For 6300, 95% of $\sigma_{day}$ are below 1.5 ppm for Ny-Ålesund and below 2 ppm for Burgos. For w4790 InGaAs, over 95% (2 standard deviations) of the error $\sigma_{day}$ are below 2 ppm for both locations. For w4790 Insb, around 95%

are below 2.75 ppm. The $\sigma_{day}$ is larger for w4790 InSb, but one thing to consider is the difference in number of data points between InSb and InGaAs that affects the standard deviation.

### 5.1   Diurnal variations

The precision of the column-averaged dry air mole fraction is estimated from its diurnal variation (**?**), see equation 5 for definition. Part of the diurnal variation (DV) is caused by the real variations in the atmospheric xCO2, dependent on the local

natural diurnal cycles, while the other part are the errors, therefore this method gives an upper limit for the precision.

**Table 3.** Estimated errors for the retrievals, the mean of the absolute value of the diurnal variation, DV; the mean standard deviation, $\sigma$; and the standard error, $\sigma/\sqrt{N}$, of the daily means.

| Location | Retrieval | DV[%] | $\sigma_{day}$[ppm] | $\sigma_{day}/\sqrt{N}$[ppm] |
|---|---|---|---|---|
| Ny-Ålesund | w4790 InGaAs | 0.182 | 1.041 | 0.167 |
|  | w4790 InSb | 0.103 | 1.472 | 0.893 |
|  | 6300 | 0.085 | 0.601 | 0.105 |
| Burgos | w4790 InGaAs | 0.214 | 1.195 | 0.128 |
|  | 6300 | 0.140 | 0.628 | 0.109 |

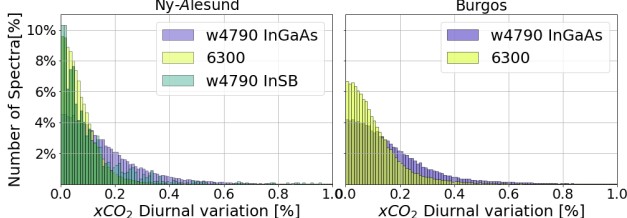

**Figure 7.** Histograms of the diurnal variations, Ny-Ålesund on the left and Burgos on the right.

The diurnal variations follow a similar pattern for all InGaAs retrievals, following a normal distribution. However, w4790 InSb looks different than the rest. For w4790 InGaAs, 95% of the diurnal variations are below 0.384% in Ny-Ålesund and 0.425% in Burgos, while for w4790 InSb 95% are below 0.246% . The fewer amount of data points measured per day for InSb might explain why the DV extends to 1%. There is a total of less than 500 data points for the three years for InSb, while there are over 4600 InGaAs data points for the same three years. For 6300, 95% of the diurnal variations are below 0.103% in Ny-Ålesund and 0.247% in Burgos. As we saw in section 5, the retrievals for w4790 have more scattering than 6300, this can be seen in the diurnal variation as well.

## 5.2 Perturbations of potential error sources

This subsection presents a sensitivity study of the w4790 xCO2 retrieval, where several retrieval input parameters were perturbed by a realistic amount following a similar method to Wunch et al. (2011a). For the InGaAs tests, Burgos spectra were used and for the InSb tests, Ny-Ålesund spectra were used. The quantities are listed in table 4 followed by the average of each of the perturbations for each site. All tests, except CO2, were performed with positive and negative perturbations (i.e. +5s and -5s), this is to evaluate if they behave in a symmetric way.

The perturbations were performed by slightly modifying variables in different retrieval steps. Two perturbations were done by multiplying, adding or subtracting from the whole profile in the a priori input files. The H2O profile was multiplied by





**Table 4.** The w4790 error budget, in the first column the error source, the second is the magnitude of the perturbation. The last two columns are the average magnitudes of the fractional difference (FD) per perturbation of the retrieved xCO2 for Ny-Ålesund and Burgos. * Corresponds to the perturbations used for the calculation of the total sum of the squared errors from these sources as similarly performed in Wunch et al. (2011a)

| Error source | Magnitude of perturbation | Ny-Ålesund $\mu FD[\%]$ | Burgos $\mu$ FD[%] |
|---|---|---|---|
| $CO_2$ profile* | constant 400ppm | 0.0971 | 0.1067 |
| $H_2O$ profile* | -5% | 0.0005 | 0.0081 |
| $H_2O$ profile | +5% | -0.0005 | -0.0077 |
| Temperature profile* | $-1°C$ | -1.5762 | -1.3501 |
| Temperature profile | $+1°C$ | 1.5438 | 1.3678 |
| Surface pressure* | -0.1% | -0.1102 | -0.1148 |
| Surface pressure | +0.1% | 0.1101 | 0.1144 |
| Time* | -10s | -0.0120 | -0.0183 |
| Time | +10s | 0.0119 | 0.0185 |
| Total $= \sum_{n=5} \mu FD*^2$ | - | 2.5061 | 1.8477 |

1.05 or 0.95 and the temperature profile $\pm 1°C$ was added. The CO2 perturbation was done by replacing the a priori profile with 400 ppm for all layers. For pressure, the measured surface pressure (*pout*) was multiplied by 1.0001 or 0.9999. Lastly, time of measurement was perturbed by adding $\pm 10$s. Then the output of each perturbed retrieval was evaluated against the unperturbed case by calculating the fractional difference (FD) with the following formula:

$$FD = 100 \times (xCO_2^{unp} - xCO_2^p)/xCO_2^{unp} \qquad (7)$$

Where $xCO_2^{unp}$ is the unperturbed total column and $xCO_2^p$ is the perturbed total column.

The w4790 retrieval is more sensitive to pressure perturbations than TCCON, which is expected, as it makes use of pressure in the calculation of the dry-air mole fraction. From Wunch et al. (2011a) we know that the largest errors for xCO2 computed using surface pressure are zero-level offsets, surface pressure errors and Sun Tracker pointing errors and that those are larger compared to using $CO_2/O_2$. This is consistent with the findings of the perturbation tests performed for tin study.

Along with pressure, as mentioned in previous sections, the temperature sensitivity of w4790 is also an important source of error. Ny-Ålesund and Burgos temperatures perturbations behave symmetrically and are the largest source of error. The negative perturbation's mean is $-1.5\%/K$ and the positive perturbation's mean is $1.5\%/K$.

The time perturbations have both, positive and negative effects depending on the time of the day. For the +10 s case, the effect is a negative bias in the morning and a positive error in the afternoon. The opposite happens in the -10 s case. Both intersect at the zero line at the minimum SZA of the day. This behaviour was expected due to the nature of the SZA and time corrections





observed in spectroscopy gas retrievals. The results of the perturbations of pressure and H2O that have both positive and negative magnitudes behave symmetrically. For pressure, a perturbation of +0.1% generates a negative difference on the xCO2

retrieval of approximately -0.1% and vice-versa, for both Burgos and Ny-Ålesund. For H2O, the smallest of all perturbations, a positive perturbation also causes a negative difference in the xCO2 retrieval. However, the magnitude of the difference, is larger in Burgos than in Ny-Ålesund. This is to be expected, given that the atmospheric water content in Burgos is larger.

For the perturbations of xCO2, we have a different effect in both locations. For Burgos, a constant a priori, causes an offset of less than 0.1%. But for Ny-Ålesund, the effect is SZA dependent, having the largest difference (-0.122%) at the largest

SZA during the morning. Then it decreases, crossing zero at 65° and at mid day, when the SZA is the smallest ($\sim$ 60°), the difference is positive ($\sim$0.05%). Finally, at the end of the day, the difference is negative once again.

The total average error caused by all the four sources is calculated by adding the square of the mean of the negative perturbations (denoted by * in the table 4), resulting in 2.5% for Ny-Ålesund and -1.8% for Burgos.

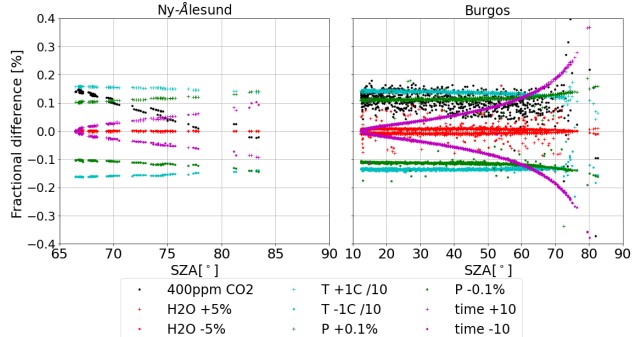

**Figure 8.** Fractional difference of perturbations (CO2, H2O, pressure, time and temperature/10) in % of the retrieved CO2 for Ny-Ålesund (InSb) on the left and Burgos (InGaAs) on the right. Remark: the values for the temperature perturbations are divided by 10 to fit in the plot range.

## 6   xCO$_2$ time series

In this section the time series is presented, for historical data in Ny-Ålesund from 1997 to 2018. There are some spectra recorded since 1992, however, the specific filter used was installed later.

The data was processed, filtered, scaled and the daily mean calculated, for all three retrievals. Comparing the w4790 xCO2 to the 6300 xCO2 for the overlapping time will show how well the seasonality is represented as well as the scattering in

comparison to the 6300 window.





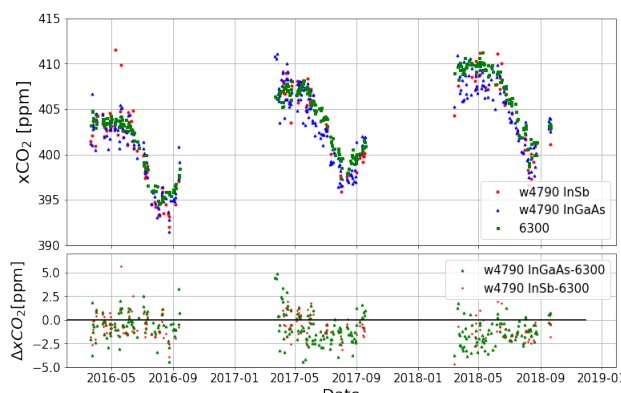

**Figure 9.** Ny-Ålesund weighted daily mean of w4790 InGaAs (in blue tringles) and w4790 InSb (in red dots) xCO2 compared with the daily means of 6300 (in green squares) xCO2 for the overlapping years. In the bottom the difference between w4790 InGaAs and 6300 (in green triangles) and Insb and 6300 (in red dots).

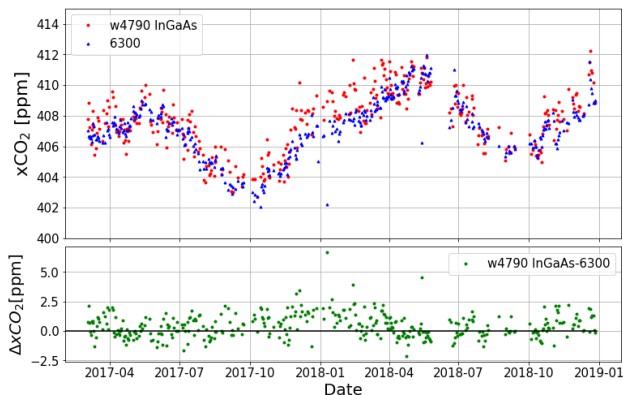

**Figure 10.** Burgos weighted daily mean of w4790 InGaAs xCO2 (in red dots) compared with the daily means of 6300 xCO2 (in blue triangles) and difference (in green) between them on the bottom

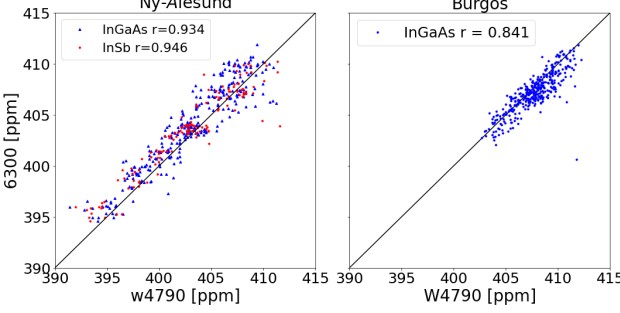

**Figure 11.** On the left, the correlation between w4790 and 6300 xCO2 for the overlapping years in Ny-Ålesund. In blue triangles w4790 InGaAs and 6300 xCO2 daily means and in red dots w4790 InSb and 6300 xCO2 daily means. On the right, the correlation between the weighted daily mean of w4790 InGaAs xCO2 and the daily means of 6300 xCO2 for Burgos.





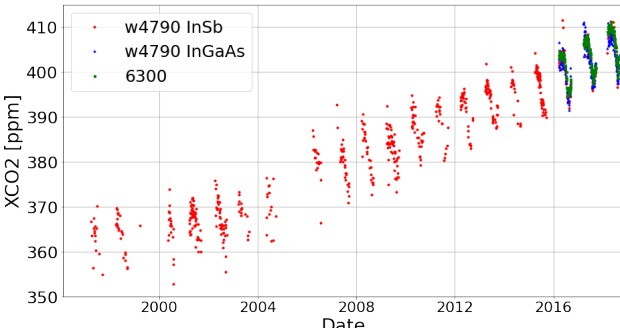

**Figure 12.** Ny-Ålesund weighted daily mean of w4790 InSb xCO2 (in red dots) for time series of all spectra available and weighted daily mean of w4790 InGaAs (in blue triangles) and the daily means of 6300 (in green squares) for the three overlapping years.

We can see that the xCO2 retrieved from w4790 can reproduce the seasonal cycle with the same amplitude of 10 ppm successfully. However, the w4790 xCO2 data points show more scattering than the 6300 xCO2, this is due to the influence of the averaging kernels as is addressed in the Appendix.

In Ny-Ålesund the recorded data covers March to October. During this period the solar zenith angle range changes. At the
peak of summer the smallest SZA is around 55° and during March and October the smallest SZA is around 73°. This means a difference in the minimum SZA of 18° Given that the maximum SZA in the datasets is 83°, the range of SZA during Spring and Autumn is quite limited. We observe in figure 9 that the difference between w4790 InGaAs and 6300 changes during the year, being higher for Spring and Autumn than for the peak of summer. Taking into account section 4.2 and the lack of air mass correction for both data sets, the changes in range of the solar zenith angle can be responsible for this observed
characteristic. The mean of the difference between w4790 InGaAs and 6300 is $-0.895$ ppm with a standard deviation of 1.513. ppm. The mean of the difference between w4790 InSb and 6300 is $-0.620$ ppm with a standard deviation of 1.516. ppm

In Burgos, where the seasonal cycle includes a dry (October to May) and a rainy season. The behaviour of its seasonality , unlike Ny-Ålesund's, can not be fitted to a cosine curve. In Burgos, like in Ny-Ålesund, there is a change in SZA depending on
the month, but larger. Around December the smallest SZA is around 40° while for June it reaches 0°, resulting in a difference in the minimum SZA of 40°, twice that of Ny-Ålesund. This means the dry season contains data with a smaller range of SZA than the wet season. We can observe that the scattering is larger during the first three months of 2018, corresponding to the dry season. Lastly, the scattering of both w4790 and 6300 is larger for the last two months of 2018 (see Fig. 10). This is similar to what is observed in Ny-Ålesund, where periods with smaller SZA range have a higher w4790 InGaAs - 6300 difference. This
can also be due to a different sensitivity towards the surface and to thin clouds. The mean of the difference between w4790 InGaAs and 6300 is 0.456 ppm with a standard deviation of 1.014 ppm.

There are several InGaAs measurements per day performed for TCCON, while there are only a couple of InSb NDACC measurements per filter per day. This influences the scattering of the InSb data points to the fitted curve. There are several





years that there is very little data points or none. In 2005 and 1999 there were technical difficulties with the instrument, therefore those years have one or no data points. . The low number of measurements per year make the InSb product very sensitive to thin clouds, and many spectra had to be filtered out due to the interferometer being abnormal.

The correlation coefficients r between the daily means of w4790 and 6300 (Fig. 11) are 0.934 for Ny-Ålesund InSb, 0.946 for Ny-Ålesund InGaAs and 0.841 for Burgos.

## 7 Retrieval strategy suggestion

Here we present recommendations on the set up for the acquisition of spectra to perform a $xCO_2$ from the $4800 \text{ cm}^{-1}$ region. There are eight standard filters used by NDACC to acquire spectra in different regions (Blumenstock et al., 2021), but there are several other filters available for use. In Ny-Ålesund the non-standard NDACC filter 4433 has been used since 1997 and in Bremen another non-standard filter, 4825, that covers 4570 to $5080 \text{ cm}^{-1}$ is used. Both filters have a good transmission in the

region of the w4790 window (see Appendix Fig. C3 and C4). For the 4433 filter the w4790 window is located near the end of the filter range but still above the 50% cut-off. Both filters have been tested for the retrieval and we can confidently recommend either.

Similarly to the protocols of NDACC, a $KBr$ or $CaF2$ beam splitter that covers the spectral range can be used for acquisition (https://ndacc.larc.nasa.gov/data/protocols/appendix-ii-infrared-ftir).

Different resolutions were tested with sample sets of InSb spectra measured in Bremen, March 2021, and used with the MIR $xCO_2$ retrieval. Lowering the resolution from $0.005 \text{ cm}^{-1}$ to $0.02 \text{ cm}^{-1}$ was found to have a negligible effect on the individual data points, while a larger number of measurements improves the precision of the retrieval and higher temporal resolution. Lower resolutions didn't show further benefit on the precision and very low resolution affects the retrieval as the spectral line shape is not resolved sufficiently. Furthermore, lower resolution allows for faster measurements therefore more measurements

which allow better performance in cloudy conditions. Given that the retrieval works just as well at a TCCON resolution of 0.02 $\text{cm}^{-1}$ (OPD 45 cm), we recommend this resolution as to increase the data points collected.

To keep errors low we suggest to use a well calibrated pressure and temperature sensor (as seen section 5.2) and use data when a smoothed xluft mean is between 0.996 and 1.002 (as seen in section 4.4). The input for the window files w4790.gnd is listed in Appendix A.

## 8 Conclusions

This study showed that it is possible to successfully retrieve the dry-air mole fraction $xCO_2$ from InSb NDACC spectra and from InGaAs TCCON spectra in the spectral window w4790. The averaging kernels show good sensitivity towards the surface. The retrieval proved to not have a big dependence on the a priori to correctly represent the daily and seasonal cycles (with correlation coefficients of at least 0.980 for all 3 retrievals. However, the w4790 InGaAs and the 6300 $xCO_2$ both have an

airmass dependence. For w4790 InGaAs, the values at larger SZA increase, while for 6300 it decreases. Additionally, w4790





retrievals are temperature dependent and will require very accurate temperature measurements. Temperature is the largest error source with $\pm\ 1.5\%/K$ followed by pressure with $\pm\ 0.11\%$ for $\pm\ 0.1\%$ error in the pressure measurement.

We showed that the xCO2 product from w4790 InGaAs is site consistent and has a scaling factor to the 6300 xCO2 product of $0.9863 \pm 0.0058$ . That scaling factor was used for both time series and the corresponding correlations. The time series shown

for the daily means show a good agreement between 6300 and w4790 (InGaAs and InSb) with a larger scattering for w4790 caused by the influence of the averaging kernels (see Appendix) and possible temperature inconsistencies.

The errors for w4790 retievals are larger than 6300 for $\sigma_{day}$ and $\frac{\sigma_{day}}{\sqrt{N}}$ but of similar magnitude for the diurnal variation (DV). In agreement with the findings by Wunch et al. (2011a), the retrieval is most sensitive to pressure perturbations, which will require to have well calibrated pressure sensors. Implementing the new suggested setup for current and new NDACC

measurement sites will provide valuable additional information on the carbon cycle.


**Appendix A: The window fitting parameters**

These are the contents of the window file w4790.gnd used as input in the GFIT retrieval. The O2 line is commented out but can be used if the spectra include it.

2 2

| | Center | Width | MIT | A | I | F | | Parameters to fit | Gases to fit |
|---|---|---|---|---|---|---|---|---|---|
| 370 | 4555.00 | 0.12 | 0 | 1 | 1 | 0 | | | : luft |
| | 6146.90 | 1.60 | 0 | 1 | 1 | 0 | | | : luft |
| | 4565.20 | 2.50 | 15 | 1 | 1 | 0 | | cl ct fs so | : h2o co2 ch4 |
| | 4570.35 | 3.10 | 15 | 1 | 1 | 0 | | cl ct fs so | : h2o co2 ch4 |
| 375 | 4571.75 | 2.50 | 15 | 1 | 1 | 0 | | cl ct fs so | : h2o co2 ch4 |
| | 4576.85 | 1.90 | 15 | 1 | 1 | 0 | | cl ct fs so | : h2o ch4 |
| | 4790.60 | 20.00 | 20 | 1 | 1 | 0 | | cl ct cc fs so ak | : co2 h2o hdo ch4 n2o |
| | : 7885.00 | 240.00 | | 15 | 1 | 1 | 0 | cl ct fs so | : o2 0o2 h2o hf co2 |





**Appendix B: The effect of the averaging kernels**

To evaluate how the averaging kernels (AK) affect the retrieval a smoothing of the 6300 retrieval with the w4790 InSb averaging kernels was done. Using the 1°AK was possible to interpolate to the exact solar zenith angle (SZA) of each of the spectra in 6300 and using Eq.(25) from Rodgers and Connor (2003) as done by Wunch et al. (2011b):

$$\hat{c}'_{12} = c_c + \sum_j h_j a_{1j}(\gamma x_c - x_c) \tag{B1}$$

where $\hat{c}'_{12}$ is the smoothed total column, InSb = 1 and 6300 = 2, $h_j$ is the pressure weighting, $a$ is the averaging kernels from the InSb spectra, $\gamma$ is the 6300 scaling factor applied to the a priori profile, $x_c$, to get the final 6300 profile that then is integrated to produce the total column $\hat{c}_2$ (Wunch et al., 2011b).

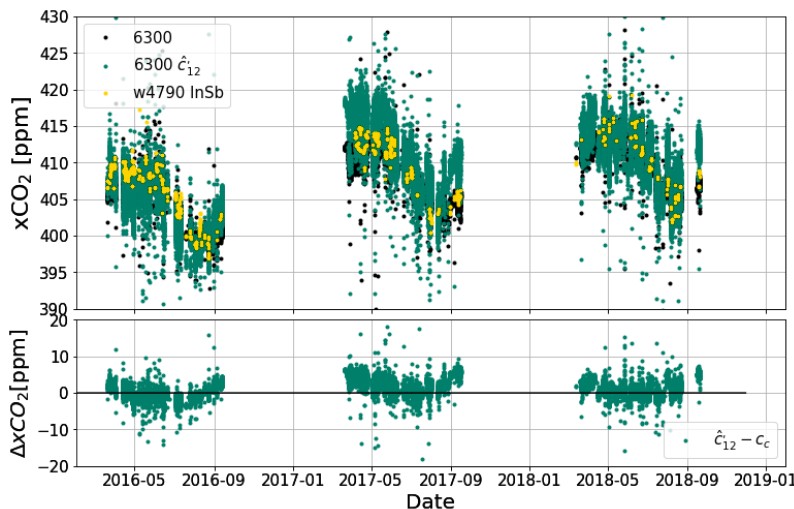

**Figure B1.** The comparison of xCO2 from 6300 in black, below the 6300 smoothed with the InSb averaging kernels in green and in yellow, the InSb w4790. Bottom panel is the difference between the 6300 smoothed $\hat{c}'_{12}$ and the 6300 xCO2 $\hat{c}_2$ .

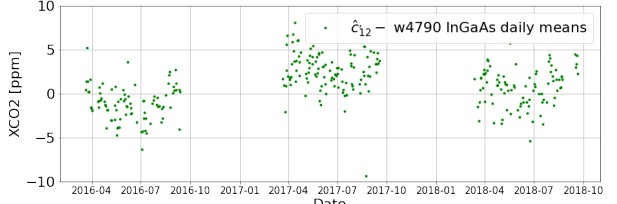

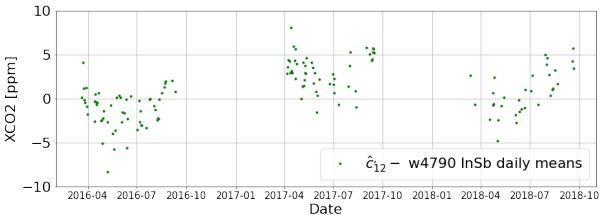

**Figure B2.** The smoothed TCCON $\hat{c}'_{12}$ daily mean minus w4790 InSb daily mean.

**Figure B3.** The smoothed TCCON $\hat{c}'_{12}$ daily mean minus w4790 InGaAs daily mean.

This comparison was chosen over using aircraft profiles due to availability, aircraft profiles are scarce, do not cover a broad range of weather conditions nor the variation with SZA. Hence a comparison with aircraft would have been very limited.





## Appendix C: The filters and their spectra

Here we show the measurement of solar spectra with each of the filter. The first is the NDACC 4433 used in Ny-Ålesund, the second is the 4750 filter used in Bremen.

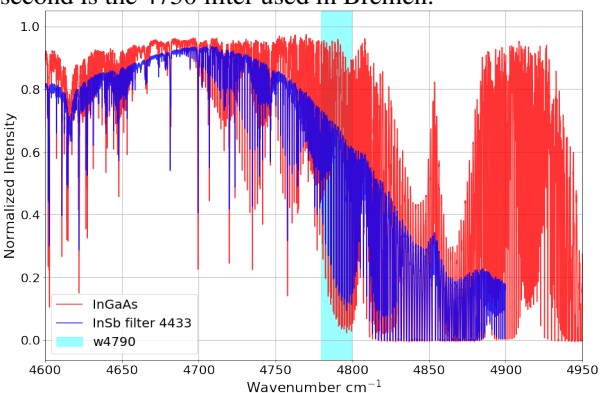

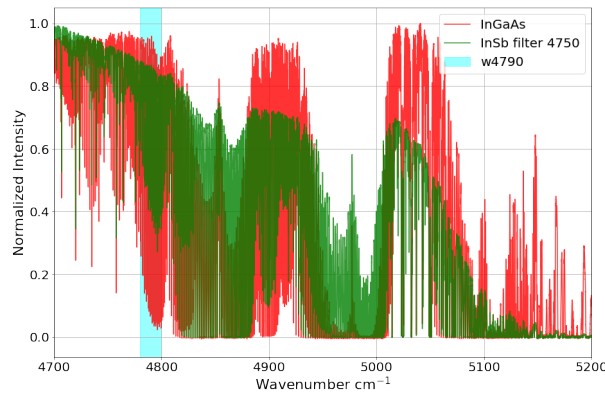


**Figure C1.** Sample solar spectra for InGaAs (in red) and InSb using the filter 4433 (in blue). Both spectra collected in Ny-Ålesund.

**Figure C2.** Sample solar spectra for InGaAs (in red) and InSb using the filter 4750 (in green). Spectra collected in Ny Ålesund and Bremen.

The transmission curves for both filters.

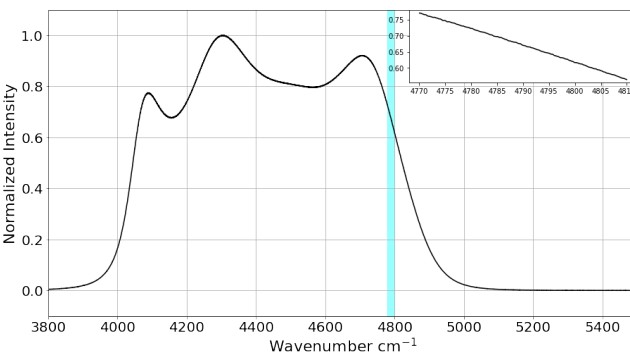

**Figure C3.** The transmission curve for the filter 4433 and a zoom in for the w4790 window.

**Figure C4.** The transmission curve for the filter 4750 and a zoom in for the w4790 window.

## Appendix D: Reduced resolution tests

In this section we show the results of the resolution tests performed in three clear days in Bremen, using the 4750 cm$^{-1}$ filter.

On the first day, four different resolutions where measured, the controls: TCCON with optical path difference (OPD) = 68.3 cm (in red), the original NDACC OPD of 180 cm (in black), then the tests, 90.01 cm, 64.3 cm, and 45.01 cm (in blue). On the





second and third day, lower resolutions where measured, 18.01 cm, 12.01 cm, 9.01 cm, 6.01 cm, 4.51 cm, 3.01 cm 2.26 cm and 1.81 cm (in green) with the corresponding controls, NDACC and TCCON.

The strategy on the first day consisted of 15 minutes of measurements of each resolution and alternate two measurements

of TCCON (in red) every 30 minutes as control. In total, four rounds of measurements were performed. The figure below is showing the mean (black dot) and the measurements (smaller color dots) and the variation of each set of measurements.

**Figure D1.** The variation on xCO2 of the resolution test by OPD for the higher resolutions tested.

The lower resolution tests, performed in two days, with more random alternations, but covering at least 15 minutes of measurements. The control TCCON measurements are performed at the beginning, in the middle and end of the other measurements.

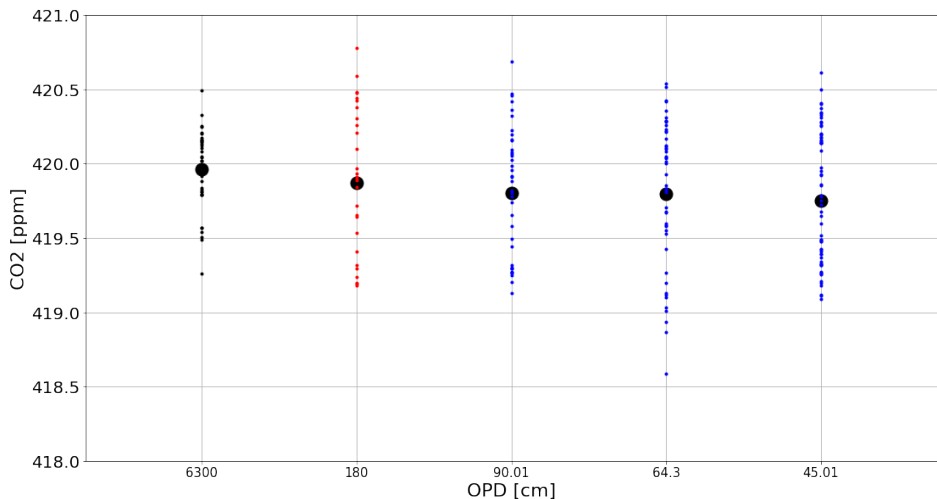


**Figure D2.** The variation on xCO2 of the resolution test by OPD for the lower resolutions tested.

In resolution Fig D1 cm there where a few outliers in the second round of measurements that were not removed. This is seen where the lower extreme in that resolution goes below the rest of tests.

The goal is to find the optimal resolution to improve the precision of the retrieval without compromising other aspects. An OPD lower than 45 cm doesn't significantly decrease the standard deviation of the grouped measurements, therefore we see no

advantage in using lower resolutions.

## Appendix E: Temperature dependence and alternative retrieval

As mentioned in section 2.2 the band chosen is a "hot" band, which makes the retrieval temperature dependent. With a lower mean energy level $E'' = 858$ cm$^{-1}$ the estimated theoretical error for a 2 K error in temperature is between 1.2 to 1.8% of the retrieved xCO2.



One alternative window considered for this retrieval was to use the third isotope 16O12C18O (named 3CO2 by HITRAN) in the same region as shown in Fig. E1, in yellow. Using the 3CO2 band would yield a less temperature dependent product, because it has a lower mean energy level, $E'' = 217 \; \mathrm{cm}^{-1}$.

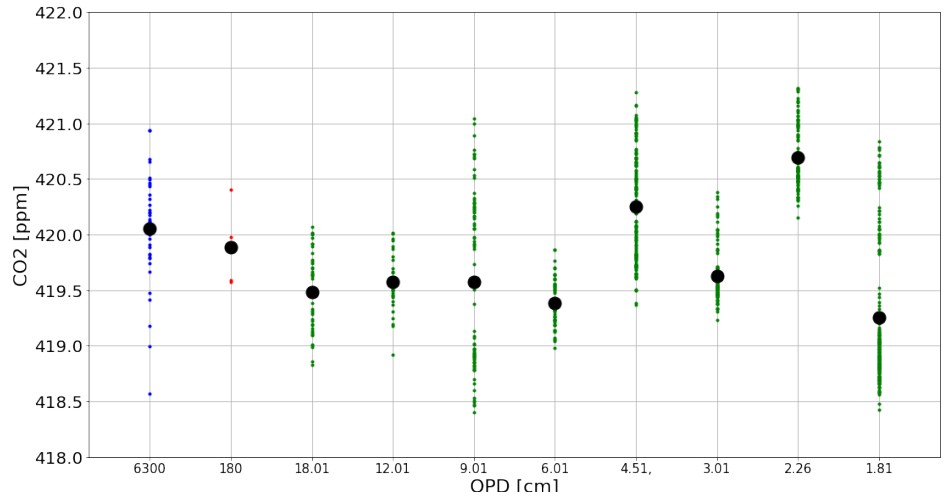

**Figure E1.** Spectra and spectral lines, in black a sample InSb spectra, in green the FTIR-atlas CO2 lines and in blue CO2. Two windows shown, in green the original w4790, and in yellow the alternative window to use 3CO2.

However, tests showed that due to the averaging kernels low sensitivity towards the surface, the retrieval doesn't capture the
seasonal cycle properly and is heavily dependent on the a priori. This alternative brings benefits as it captures the stratosphere, however there is an offset to be calculated that is possibly due to GFIT's assumed isotopic fractionation, which can be easily corrected for.





**Appendix F: The line list in w4790**

**Table F1.** The mid-IR line list used in this study. Each line is characterized by the center wavenumber. Additionally, the corresponding molecular transition is given. Data taken from the TCON GGG2020 linelist, adapted from HITRAN (Rothman et al., 2009).

| Center [cm$^{-1}$] | Line intensity [cm$^{-1}$(moleculecm$^{-2}$)] | Quantum Transition |
|---|---|---|
| 4780.99 | $5.583\times10^{-24}$ | 2 1 1 13 → 0 1 1 01 |
| 4782.86 | $6.534\times10^{-24}$ | 2 1 1 13 → 0 1 1 01 |
| 4784.70 | $7.498\times10^{-24}$ | 2 1 1 13 → 0 1 1 01 |
| 4786.15 | $8.615\times10^{-24}$ | 2 1 1 13 → 0 1 1 01 |
| 4786.53 | $8.431\times10^{-24}$ | 2 1 1 13 → 0 1 1 01 |
| 4787.90 | $9.616\times10^{-24}$ | 2 1 1 13 → 0 1 1 01 |
| 4789.63 | $1.050\times10^{-23}$ | 2 1 1 13 → 0 1 1 01 |
| 4790.12 | $9.994\times10^{-24}$ | 2 1 1 13 → 0 1 1 01 |
| 4791.35 | $1.120\times10^{-23}$ | 2 1 1 13 → 0 1 1 01 |
| 4791.89 | $1.051\times10^{-23}$ | 2 1 1 13 → 0 1 1 01 |
| 4793.05 | $1.166\times10^{-23}$ | 2 1 1 13 → 0 1 1 01 |
| 4793.64 | $1.079\times10^{-23}$ | 2 1 1 13 → 0 1 1 01 |
| 4794.42 | $6.197\times10^{-25}$ | 3 0 0 14 → 1 0 0 02 |
| 4794.73 | $1.180\times10^{-23}$ | 2 1 1 13 → 0 1 1 01 |
| 4795.36 | $1.078\times10^{-23}$ | 2 1 1 13 → 0 1 1 01 |
| 4796.40 | $1.158\times10^{-23}$ | 2 1 1 13 → 0 1 1 01 |
| 4797.05 | $1.193\times10^{-24}$ | 2 0 0 13 → 0 0 0 01 |
| 4797.07 | $1.044\times10^{-23}$ | 2 1 1 13 → 0 1 1 01 |
| 4797.45 | $1.015\times10^{-24}$ | 3 0 0 14 → 1 0 0 02 |
| 4798.06 | $1.097\times10^{-23}$ | 2 1 1 13 → 0 1 1 01 |
| 4798.77 | $9.756\times10^{-24}$ | 2 1 1 13 → 0 1 1 01 |
| 4798.95 | $1.158\times10^{-24}$ | 3 0 0 14 → 1 0 0 02 |
| 4799.17 | $1.817\times10^{-24}$ | 2 0 0 13 → 0 0 0 01 |
| 4799.70 | $9.933\times10^{-24}$ | 2 1 1 13 → 0 1 1 01 |
| 4800.43 | $1.258\times10^{-24}$ | 3 0 0 14 → 1 0 0 02 |
| 4800.44 | $8.634\times10^{-24}$ | 2 1 1 13 → 0 1 1 01 |





**Appendix G: TCCON's LCO$_2$ window**

TCCON is working on a new window in the same region centred at 4852.87 cm$^{-1}$ and of 86.26 cm$^{-1}$ width that uses the same band to retrieve CO2. The LCO$_2$ window, was not developed with the same purposes w4790 was developed, therefore, it is not ideal for retrieving historical NDACC spectra from Ny-Ålesund as done in this paper because the window is located pass the 50% cut off the 4433 filter used there. However the window would work with TCCON and NDACC spectra with other filters such as 4750 shown in Appendix C.

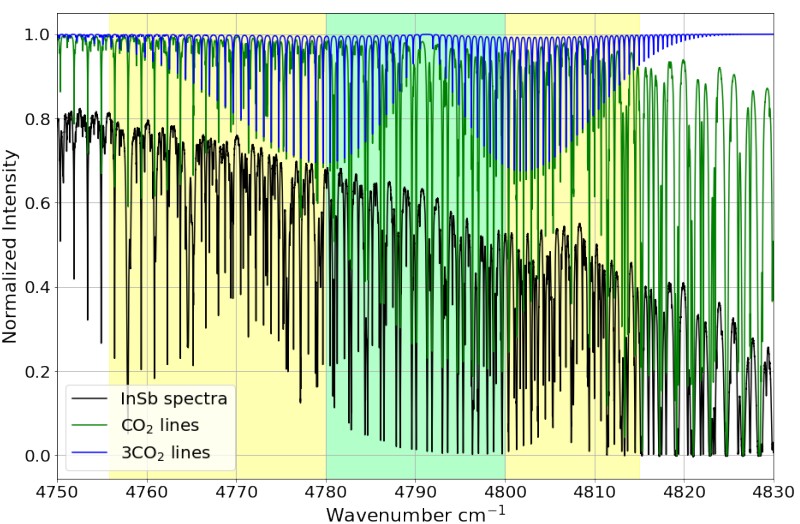

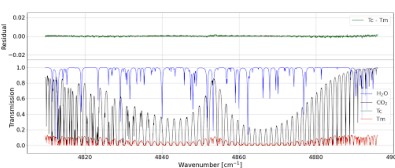

**Figure G1.** A example of the calculated spectra (Tc) and the measured spectra (Tm) for the LCO$_2$ window, CO$_2$ absorption lines, other gases and the residual (Tm − Tc)

for a typical spectra recorded in Ny-Ålesund.

**Figure G2.** Sample solar spectra for In-GaAs (in red) and InSb using the filter 443 (in blue) and filter 4750 (in green), collected in Ny-Ålesund. LCO$_2$ window in magenta.

Tht LCO$_2$ and the W4970 windows have a large difference in the averaging kernels. LCO$_2$ AKs have little variation with SZA. For low SZA w4790 is closest to 1 and the largest difference between both windows.

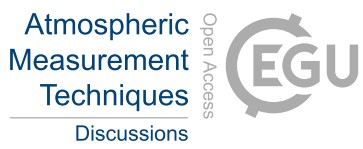

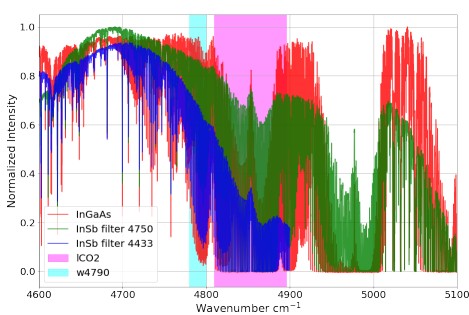

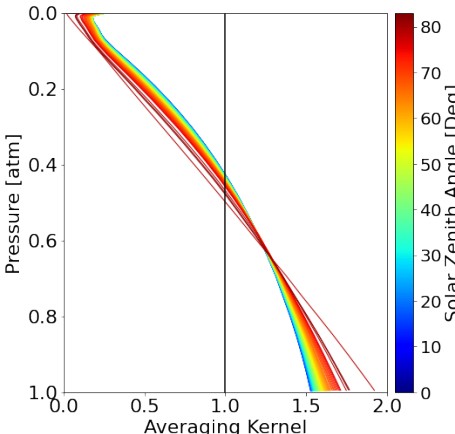

**Figure G3.** w4790 averaging kernels averaged to 1° for InGaAs Ny-Ålesund.

**Figure G4.** LCO$_2$ averaging kernels not averaged InGaAs Ny-Ålesund.





## Appendix H: Supplemental plots

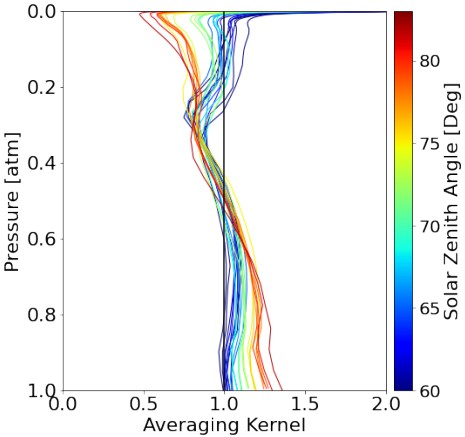

**Figure H1.** w4790 averaging kernels averaged to
1° for InSb Ny-Ålesund spectra.

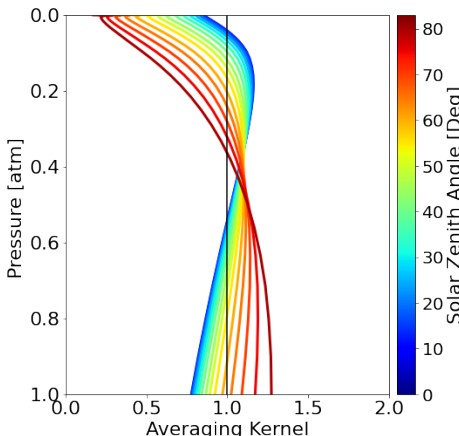

**Figure H2.** The averaging kernels for the Burgos
TCCON product (from ggg2014).

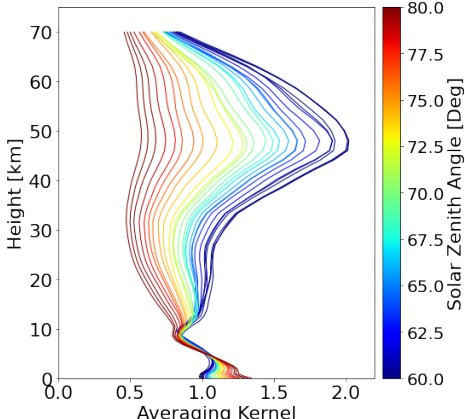

**Figure H3.** w4790 averaging kernels averaged to
1° for InGaAs Ny-Ålesund spectra plotted against
Altitude in Km.

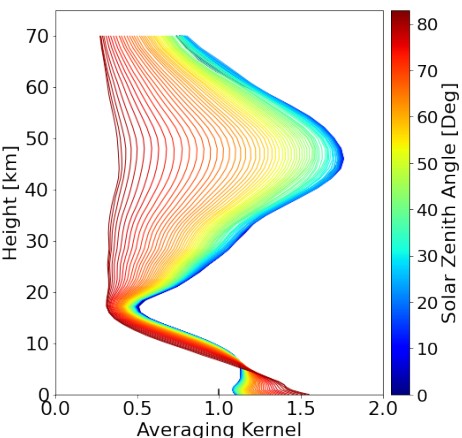

**Figure H4.** The averaging kernels for the Burgos
InGaAs spectra plotted against Altitude in Km).





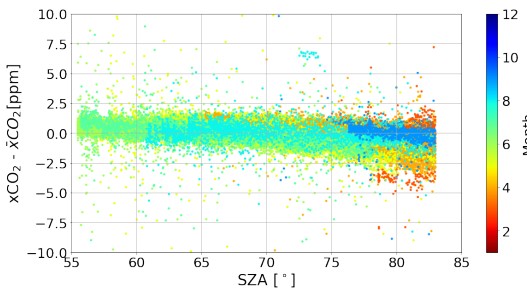

**Figure H5.** The standard and deseasonalized a priori profiles for Ny-Ålesund.

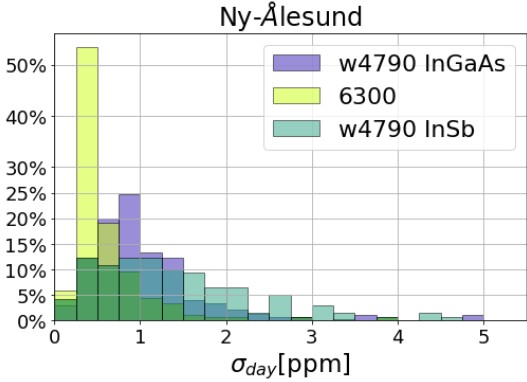

**Figure H6.** Ny-Ålesund 6300 CO2, prior to the airmass correction, minus the daily mean of the corresponding month/

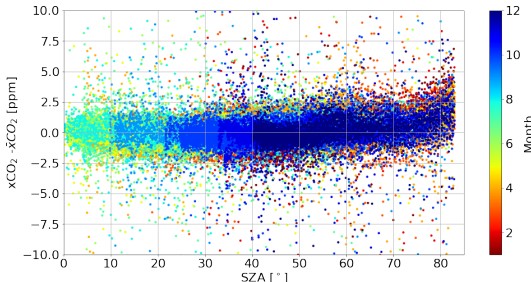

**Figure H7.** Burgos 6300 $xCO_2$, prior to the airmass correction, minus the daily mean of the corresponding month.

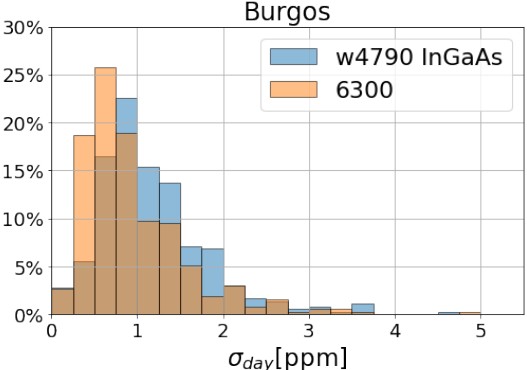

**Figure H8.** Histogram of the standard deviation of the daily means of CO2 in Ny-Ålesund.

**Figure H9.** Histogram of the standard deviation of the daily means of CO2 in Burgos.



*Author contributions.* RC determined the window, retrieved the data for Ny-Ålesun and Burgos, performed the a priori, perturbation and resolution tests, the site to site consistency and the error budget evaluation and wrote the manuscript. JL provided the in situ comparison of section 4.4. MB assisted RC closely through the process in the retrieval and post processing. TW and JN guided RC the content of the research. CP assisted RC in the retrieval using GFIT. All authors contributed to the final version of the manuscript.

*Competing interests.* One authors is a member of the editorial board of journal Atmospheric Measurement Techniques. The peer-review process was guided by an independent editor, and the authors have also no other competing interests to declare.

*Acknowledgements.* A portion of this research was carried out at the Jet Propulsion Laboratory, California Institute of Technology, under a contract with the National Aeronautics and Space Administration (80NM0018D0004).

The TCCON station in Burgos is supported in part by the GOSAT series project. Local support for Burgos is provided by the Energy Development Corporation (EDC, Philippines).





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
