# Peer review of "A retrieval of xCO2 from ground-based mid-infrared NDACC solar absorption spectra and comparison to TCCON"

_Atmospheric Measurement Techniques, 2023_

## Referee Comment (RC1)

**Review of A retrieval of xCO$_2$ from ground-based mid-infrared NDACC solar absorption spectra and comparison to TCCON" (AMT-2023-32)**

The manuscript introduces a new spectral window to retrieve total column abundances of CO2 (XCO$_2$) in the mid-infrared region measured by the NDACC network. Comparisons are made with standard TCCON XCO$_2$ product as well as the retrieved values from the same spectral window in TCCON. This study is valuable since it adds a new data product to NDACC that could potentially expand the total column CO$_2$ measurement network around the globe.

The formatting of the manuscript, figures and tables need to be improved for better presentation of the content. Some passages in the text require extra explanation to help the audience understand the ideas more clearly. Here are some general comments that I think would help improve the manuscript:

1. Please consider making the figures, axis labels and numbers, legends, symbols larger so they are easier to read.

2. For CO$_2$, O$_2$ , N$_2$, CH$_4$ , use subscripts throughout the text.

3. There are undefined citations throughout the text shown as question marks. Please make sure to link them to the right references.

4. Define all the variables introduced in the equations before or after the equation.

5. In section 4.2, why don't you apply an airmass correction to the retrievals to see if it improves the retrievals? and maybe use those for comparisons in the following sections?

6. What is the purpose of section 5.1? If it's to evaluate the precision of measurements why don't you use a shorter period of time let's say 20 minutes for averaging. Because as you said we expect XCO$_2$ to change during the day and the SD wouldn't necessarily represent the precision of measurements. If the reason is the limited number of measurement points from NDACC please state it in the text.

7. In table 4, could you add a column presenting the values from Wunch et al to make the comparisons easier?

Minor corrections and comments:

- **Line 10:** Do you mean precision **of** 0.2%?

- **Line 10:** I am assuming in the first part of the sentence you are talking about the averaging kernel comparisons and at the end you are talking about the seasonality observed in the retrieved value? Can you make this more clear? Maybe split it up into two sentences?

- **Line 11:** We don't usually refer to specific sections in the abstract. Maybe you can say: In addition, we propose an optimal retrieval strategy to improve the quality of the data product.

- **Line 15:** Define $XCO_2$ for the first time.

- **Line 15:** main net sources of what? $CO_2$?

- **Line 20:** validation is repeated twice

- **Line 20:** need additional space after of $CO_2$

- **Line 25:** do you mean pre recorded spectra?

- **Line 27:** NDACC trace gas products, ...

- **Line 28:** Expand the temporal and spatial coverage of the total column products.

- **Line 30:** Define $XCO_2$ for the first time.

- **Line 42:** What do you mean by 'topospheric signal is damped in comparison to TCCON' ?

- **Line 60:** ... since 1992 which covers the mid-infrared ...

- **Line 82:** 21113 $\rightarrow$ 01101 Add more explanation for this notation or remove it here and refer to table F1.

- **Line 85:** Buckingham (1976) should be in brackets.

- **Line 90:** Which appendix?

- **Line 92:** 'on the left' / 'towards the left' : please be more specific. You can use the approximate wavenumber.

- **Line 102:** Please move this to the data availability section at the end of the manuscript.

- **Line 108 :** Wunch et al. (2015) is describing GGG2014. GGG2020 is still in preparation you can cite https://agu.confex.com/agu/fm20/meetingapp.cgi/Paper/675531 for now.

- **Equation 1:** Define VC here.

- **Line 125:** These are two main important differences...

- **Line 126:** you haven't explained what is airmass and what is in situ correction yet.

- **Line 132:** not abundant (usually less than 20) Be more clear. What do you mean by less than 20? less than 20 spectra per day?

- **Equations 3 and 4:** Define all the elements in the equations.

- **Line 141:** This sentence is not clear.

- **Line 143:** Which retrieval windows are the test spectra from?

- **Line 151:** Strong sensitivity/ high sensitivity instead of good sensitivity.

- **Line 156:** ... follow a similar curvature but at different altitude ... Not clear. Please explain in more details.

- **Line 167:** Add citation to GEOS-FO-IT.

- **Line 171:** Unclear. Break down the sentence into two or three. Explain how you construct the modified a priori separately in another sentence. Also what do you mean by other tests?

- **Line 174:** same retrieval: do you mean same retrieval procedure or same retrieval algorithm?

- **Figure 3:** Define $\Delta CO_2$ both in the plot and in the caption. Is it fixed a priori minus standard a priori or vice versa?

- **Line 177:** correlation coefficient r (between which two parameters?)

- **Line 179:** Please be consistent with Figure 3. $\delta$ $CO_2$ is the same as $\Delta$ $CO_2$?

- **Line 179:** What do you mean by modified apriori? Is it the same as what you call as the fixed a priori used in Figure 3.

- **Line 181:** a over estimation $\rightarrow$an over estimation

- **Line 183:** I don't think artifact is a good description for the airmass independent correction. Airmass independent bias might be better.

- **Line 185:** Add brackets to Wunch et al. (2011a)

- **Line 187:** TCCON derives and applies a single empirical airmass correction: It's not in fact a single value. GGG2020 applies airmass correction for each retrieval window separately before averaging them for each gas.

- **Line 192:** the dependence is observed: which dependence are you referring to?

- **Figure 4:** add : after Top in the caption.

- **Line 204:** To investigate if the scaling between 6300 and w4790 is consistent between both locations....

- **Figure 5:** Define the ratio in the caption. Also what do you mean by $H_2O$ content? Is that $XH_2O$? In addition, readers with colour vision deficiencies might not be able to differentiate between these colors. Please double check.

- **Line 216:** No brackets for Wunch et al.

- **Line 218:** A higher dependence in what? How do you define $XCO_2$ bias? Do you mean bias between $XCO_2$ and column averaged in situ profiles?

- **Line 225:** ... , that aircrafts and balloons can't capture, **in addition** to the temperature sensitivity of the spectral window.

- **Line 228:** The mean ratio... specify which ratio you are referring to.

- **Line 229:** These bias**es** or **This** bias

- **Line 229:** airmass correction factor: do you mean in-situ correction factor?

- **Line 239:** are you using 1 standard deviation for 6300 and 2 for 4790? why?

- **Line 240:** why don't you use standard error instead then?

- **Figure 7:** maybe bring the purple histogram forward for better visibility.

- **Line 258:** All tests, except $CO_2$,... Do you mean $CO_2$ a priri profile?

- **Line 268:** The w4790 retrieval is more sensitive to pressure perturbations than TCCON. How much is the value for TCCON?

- **Line 271:** ... the perturbation tests performed for this study.

- **Line 283:** For the perturbations of xCO2 ... Again do you mean perturbations in the a priori profile?

- **Figure 8:** Again here I guess you mean $CO_2$ and $H_2O$ a priori.

- **Line 306:** in the minimum SZA of 18∘ . Given that the maximum SZA ...

- **Line 326:** Could you elaborate more on what you mean by the interferometer being abnormal?

- **Line 391:** Add proper citation.

- **Line 347:** how would a good temperature sensor improve errors?

- **Line 328** explain what is a window file.

- **Line 352:** good sensitivity... high sensitivity?

- **Line 354:** where does the bracket close?

- **Appendix A:** Explain the parameters in the table.

- **Line 382:** What do you mean by 1∘AK? Do you mean averaging kernels were binned by 1 degree SZAs?

- **Line 393:** ... each of the filter**s**.

- **Line 403:** refer to the corresponding figures in this sentence and the previous sentence.

- **Figure D2:** is 6300 opd in cm? Also could you add a legend describing colors? Also mention the date and time of measurements in the caption.

- **Line 411:** In resolution Fig D1 cm... not clear what you are referring to here!

- **Figure E1:** this caption doesn't look like describing this figure.

- **Table F1:** Describe the vibrational and rotational levels corresponding to each digit.

- **Figure G3:** caption doesn't match the figure.

- **Figure H4:** InGaAs spectra: which spectral window?

- **Figure H5:** which one is on top and which one at the bottom?

---

## Referee Comment (RC2)

General comments:

The authors present a study on the retrieval of $XCO_2$ from NDACC MIR spectra. Up to now, $CO_2$ is not a target gas within NDACC. A $CO_2$ retrieval would further exploit MIR spectra recorded since a long time and would nicely extend the capabilities of the NDACC network.

The authors compared these $XCO_2$ data with TCCON $XCO_2$ data and characterized the error sources of this $XCO_2$ data product. The precision of the retrieved $XCO_2$ is sufficiently high to create a useful data set. Some details, however, like the instrumental line shape of the spectrometer or the accuracy of the pressure sensor that might affect the long-term stability of the retrieved $XCO_2$ data need more discussion.

Therefore, I would recommend publishing this paper after some revisions. Please also see specific comments below. The paper is well structured and fits well to the scope of AMT.

Specific comments:

The optical filter used for this study differ from the NDACC standard filter set. NDACC filter #1 covers the region from 3850–4550 $cm^{-1}$ and does not include the 4800 $cm^{-1}$ micro window. The filter used in this study is optional and in NDACC numbering (increasing number with increasing wavelength) something like NDACC filter #0. Therefore, it is questionable whether many sites recorded spectra with it since a long time.

Since the MIR does not include $O_2$ signatures pressure data have been used to calculate $XCO_2$. Accordingly, the quality of the $XCO_2$ product relies on the quality of the pressure sensor and its data. The paper would benefit from a paragraph or appendix describing the used pressure sensor and its requirements. In particular, the required precision and accuracy and any means to check and correct for drifts need to be discussed in the paper.

The paper includes an error estimate and the precision of the retrieved $XCO_2$ product is good. Normally, a correlation coefficient of 0.95 (Fig. 11) is quite good, too. For $XCO_2$ data however, it is not fully clear whether this is sufficient. For Burgos site, the correlation coefficient (0.84) is even lower.

It is certainly a good idea to include spectra from a wet site. The site used for this, however, suffers from the lack of NDACC spectra. Instead, spectra were recorded with a broadband InGaAs detector without any optical filter. In the 4800 $cm^{-1}$ region an NDACC type spectrum recorded with a cooled InSb detector and bandwidth filter is better with respect to linearity and signal to noise ratio.

You might add the signal to noise ratio of the InSb and InGaAs spectra as shown in Figs. C1&C2. This might give a hint on the noise level although the spectral resolution differs strongly. Alternatively, did you record these spectra (C1&C2) with the same spectral resolution?

The spectral resolution of 0.005 $cm^{-1}$ as used in the NDACC mode is probably overdone in this spectral regime at 4800 $cm^{-1}$. (At this spectral regime, my personnel guess would be something about 90 cm OPD or a resolution of about 0.01 $cm^{-1}$, respectively.) The authors tested different spectral resolutions (Appendix D) which is a very useful exercise. However, I do not see a clear

statement on the optimal resolution at 4800 cm$^{-1}$. I would suggest adding a plot of a $CO_2$ line recorded with different OPD to study the needed OPD to resolve the line fully.

In the introduction, the paper states that previously published recipes for $XCO_2$ retrievals does not yield $XCO_2$ data of sufficient quality to use the data for atmospheric research. This is correct. However, in the discussion or conclusion the results of the data set retrieved in this paper have not been compared to those previous data sets to demonstrate the improved data quality.

Furthermore, the long-term stability of a time series of this data product is not studied in detail. This should include a discussion of a possible drift of the pressure sensor. See comment above. Secondly, regular cell measurements to retrieve the ILS (Instrumental Line Shape) of the spectrometer are strongly recommended over the entire time series. Are ILS measurements made and used in this study and if so what are the ILS results? ILS parameters are also missing in the error estimate (Table 4).

While chapter 6 announces historical data from 1997 to 2018, Figs. 9, 10 and B1 just show data for the years 2016, 2017 and 2018. It seems this is limited to the availability of data from the 6300 cm$^{-1}$ spectral region, isn't it? The NDACC type measurements at 4800 cm$^{-1}$ started in 1997. When did the TCCON measurements at 6300 cm$^{-1}$ started at Ny-Ålesund? It would be helpful for the reader if you add a table or columns to Table 1 listing the periods of available and used spectra for each retrieval and chapter. Also for Table 3 and Fig.7 the period covered would be helpful.

Finally, if earlier TCCON data (from before 2016) are available it would be nice to include a longer data set into the comparison. Moreover, to study the long-term stability in more detail it would be good to calculate the difference of NDACC and TCCON $XCO_2$ data from the beginning of these measurements and to perform a trend study on the difference.

Technical corrections:

- l. 20: validation validation
- l. 20: of$CO_2$
- l. 25: sppectru, => spectrum
- l. 59: In NDACC spectra => NDACC spectra
- l. 126: Correct use of respectively?
- l. 141 and a few more times there is still a question mark
- l. 158: Ny-Ålesund around => Ny-Ålesund and around
- l. 163: following => following equation
- l. 169: when => even if?
- l. 185: => (Wunch et al., 2011a)
- l. 226: the those => those?
- l. 229: These bias => These biases or this bias
- l. 311: is and full stop are missing.
- l. 326: points. . => points.
- l. 407: Fig. D1 is missing or the number of Fig. D2 is incorrect.
- l. 412: where => were or better rephrase the sentence beginning with 'In resolution Fig D1'
- l. 445: Ny-Ålesun => Ny-Ålesund
- l. 449: One authors => One author

The list of typos is not complete. Please proofread carefully.

---

## Author Comment (AC1)

**improvementResponse to reviewer 1.**

The authors appreciate the comments and corrections of the reviewers.

The formatting of the manuscript, figures and tables need to be improved for better presentation of the content. Some passages in the text require extra explanation to help the audience understand the ideas more clearly. Here are some general comments that I think would help improve the manuscript:

1. Please consider making the figures, axis labels and numbers, legends, symbols larger so they are easier to read. Done.

2. For $CO_2$ , $O_2$ , $N_2$ , $CH_4$ , use subscripts throughout the text. Done.

3. There are undefined citations throughout the text shown as question marks. Please make sure to link them to the right references. Corrected.

4. Define all the variables introduced in the equations before or after the Equation. Done.

5. In section 4.2, why don't you apply an airmass correction to the retrievals to see if it improves the retrievals? and maybe use those for comparisons in the following sections?
 In section 4.2 there is no air mass correction because the test is to determine if one is needed. The correction is calculated in section 4.3 and not from the airmass directly because we wanted to see how the retrieval without modification behaves against 6300.

6. What is the purpose of section 5.1? If it's to evaluate the precision of measurements why don't you use a shorter period of time let's say 20 minutes for averaging. Because as you said we expect $XCO_2$ to change during the day and the SD wouldn't necessarily represent the precision of measurements. If the reason is the limited number of measurement points from NDACC please state it in the text.
It is indeed due to the lack of measurements from InSb. One day is the minimum to average a few data points. This has been included in the text.
(" **However this strategy was still chosen as, due to the small amount of data points of InSb specta, it is the shorter time period to make a meaningful average**." line 276)

7. In table 4, could you add a column presenting the values from Wunch et al to make the comparisons easier? Adding those values would not result in an equivalent comparison. In Wunch et. al, the values used are at SZA  20 and 70. However, because of limited data availability for those tests (2 days) the average of all SZA were used. This difference makes a proper comparison harder, however the plot could be added in the supplemental plots for a comparison of the behaviour of the perturbations.

Minor corrections and comments:
• Line 10: Do you mean precision of 0.2%? Corrected.
• Line 10: I am assuming in the first part of the sentence you are talking about the averaging kernel comparisons and at the end you are talking about the seasonality observed in the

retrieved value? Can you make this more clear? Maybe split it up into two sentences?
Corrected.

• Line 11: We don't usually refer to specific sections in the abstract. Maybe you can say: In addition, we propose an optimal retrieval strategy to improve the quality of the data product. Corrected.

• Line 15: Define XCO 2 for the first time. Corrected.

• Line 15: main net sources of what? CO 2 ? Corrected.

• Line 20: validation is repeated twice Corrected.

• Line 20: need additional space after of CO 2 Corrected.

• Line 25: do you mean pre recorded spectra? No, per or for each, I clarified this. Line 28

• Line 27: NDACC trace gas products, ... Corrected.

• Line 28: Expand the temporal and spatial coverage of the total column products. Corrected.

• Line 30: Define XCO 2 for the first time. Corrected.

• Line 42: What do you mean by 'tropospheric signal is damped in comparison to TCCON' ? lower sensitivity due to the low AKs, I used the same language as in the reference. It is corrected and clarified. Line 46

• Line 60: ... since 1992 which covers the mid-infrared ...Corrected.

• Line 82: 21113 → 01101 Add more explanation for this notation or remove it here and refer to table F1. $\nu_1 \nu_2 l_2 \nu_3 n$ Three of these, $\nu_1\nu_2\nu_3$, express the number of quanta activated for each fundamental; $l2$ is the $l$ value for the degenerate $\nu_2$ fundamental and its overtones; the fifth integer is the $n$th component of the Fermi interacting $\nu_1$ and $2\nu_2$ vibrational states including their overtone and combination states). (Toth et. al, 2008) . Line 87

• Line 85: Buckingham (1976) should be in brackets.Corrected.

• Line 90: Which appendix? Corrected.

• Line 92: 'on the left' / 'towards the left' : please be more specific. You can use the approximate wavenumber. Corrected.

• Line 102: Please move this to the data availability section at the end of the manuscript. Corrected.

• Line 108 : Wunch et al. (2015) is describing GGG2014. GGG2020 is still in preparation you can cite https://agu.confex.com/agu/fm20/meetingapp.cgi/Paper/675531 for now. Corrected.

• Equation 1: Define VC here. Corrected.

• Line 125: These are two main important differences...Corrected.

• Line 126: you haven't explained what is airmass and what is in situ correction yet. No, the airmass or in situ corrections are described in section 4.2 and 4.4 correspondingly, with more detail. This is now stated in the text. Line 137

• Line 132: not abundant (usually less than 20) Be more clear. What do you mean by less than 20? less than 20 spectra per day? Yes less than 20. I clarified.

• Equations 3 and 4: Define all the elements in the equations. Corrected.

• Line 141: This sentence is not clear. Corrected.

• Line 143: Which retrieval windows are the test spectra from? The question is not clear. If you mean which spectra the retrievals from I have corrected the text.

• Line 151: Strong sensitivity/ high sensitivity instead of good sensitivity. Corrected.

• Line 156: ... follow a similar curvature but at different altitude ... Not clear. Please explain in more details. Done.

• Line 167: Add citation to GEOS-FO-IT. Done.
https://gmao.gsfc.nasa.gov/pubs/docs/Lucchesi865.pdf  Line 184

• Line 171: Unclear. Break down the sentence into two or three. Explain how you construct the modified a priori separately in another sentence. Also what do you mean by other tests? Corrected.

• Line 174: same retrieval: do you mean same retrieval procedure or same retrieval algorithm? Both. It has being clarified in line 192

• Figure 3: Define $\Delta CO_2$ both in the plot and in the caption. Is it fixed a priori minus standard a priori or vice versa? I defined it in the caption as it would not fit in the Y axis of the plot.

• Line 177: correlation coefficient r (between which two parameters?) The retrievals with the fixed and standard a priori. Clarified.  Line 195

• Line 179: Please be consistent with Figure 3. $\delta CO_2$ is the same as $\Delta CO_2$ ? Corrected.

• Line 179: What do you mean by modified a apriori? Is it the same as what you call as the fixed a priori used in Figure 3. Yes, corrected.

• Line 181: a over estimation →an over estimation Corrected.

• Line 183: I don't think artifact is a good description for the airmass independent correction. Airmass independent bias might be better. Bias is a good alternative. I am using the language used in Wunch et al (2011a) to describe it for consistency.

• Line 185: Add brackets to Wunch et al. (2011a) Corrected.

• Line 187: TCCON derives and applies a single empirical airmass correction: It's not in fact a single value. GGG2020 applies airmass correction for each retrieval window separately before averaging them for each gas. It is a single correction, not a single value. This means the correction is only done once. This correction is performed after averaging all windows per gas, not before, it is one of the last steps of the retrieval algorithm. It is also possible to use the product of the averaged windows without the airmass correction.  This hs being rephrased for clarity

**"To correct this, TCCON applies an empirical correction. In GGG2014, the correction was derived separately for each xGAS product. In GGG2020, it is derived for each retrieval window."**  Line 206

• Line 192: the dependence is observed: which dependence are you referring to? To the SZA. Clarified.

• Figure 4: add : after Top in the caption. Corrected.

• Line 204: To investigate if the scaling between 6300 and w4790 is consistent between both locations.... Corrected.

• Figure 5: Define the ratio in the caption. Also what do you mean by H 2 O content? Is that XH 2 O? In addition, readers with colour vision deficiencies might not be able to differentiate between these colors. Please double check. Corrected. All images have been checked several times for colorblindness. The colours are different levels, so even in scales of grey they are different shades.

• Line 216: No brackets for Wunch et al. Corrected.

• Line 218: A higher dependence in what? How do you define XCO 2 bias? Do you mean bias between XCO 2 and column averaged in situ profiles? To the airmass or xluft. A systematic distortion against a reference.

• Line 225: ... , that aircrafts and balloons can't capture, in addition to the temperature sensitivity of the spectral window. Corrected.

• Line 228: The mean ratio... specify which ratio you are referring to. Corrected.

• Line 229: These biases or This bias Corrected.

• Line 229: airmass correction factor: do you mean in-situ correction factor? Clarified.

• Line 239: are you using 1 standard deviation for 6300 and 2 for 4790? Why? I use 2 standard deviations (95%) for both.

• Line 240: why don't you use standard error instead then? Corrected by adding some text and referring to table 3 where the SEM is listed.

" one thing to consider is the difference in number of data points between InSb and InGaAs that affects the standard deviation. **For this reason the standard error was also calculated (shown in table 3) to have a representation of mean erro**r." Line 270

• Figure 7: maybe bring the purple histogram forward for better visibility. Improved.

     • Line 258: All tests, except CO 2 ,... Do you mean CO 2 a priri profile? Yes. Clarified.

• Line 268: The w4790 retrieval is more sensitive to pressure perturbations than TCCON. How much is the value for TCCON? For TCCON the pressure error (-0.1% profile perturbation) is -0.036 at SZA 20° and −0.033 at SZA 70°.

• Line 271: ... the perturbation tests performed for this study. Corrected.

• Line 283: For the perturbations of xCO2 ... Again do you mean perturbations in the a priori profile? Corrected.

• Figure 8: Again here I guess you mean CO 2 and H 2 O a priori. Corrected.

• Line 306: in the minimum SZA of 18∘ . Given that the maximum SZA ... Corrected.

• Line 326: Could you elaborate more on what you mean by the interferometer being abnormal? Corrected. Of bad quality, that indicates change in source brightness like for example due to thin cloud conditions or an error in the solar tracker.

• Line 391: Add proper citation. It is unclear, citation on the lack of data?. The 67 matches shown in figure 6 is the total number available.

• Line 347: how would a good temperature sensor improve errors? Corrected. A better understanding of the temperature profile of the atmosphere to produce a more accurate a priori temperature profile. Line 360

• Line 328 explain what is a window file. Done.

• Line 352: good sensitivity... high sensitivity? Yes. Corrected.

• Line 354: where does the bracket close? Corrected.

• Appendix A: Explain the parameters in the table. Done.

• Line 382: What do you mean by 1∘AK? Do you mean averaging kernels were binned by 1 degree SZAs? Yes, Clarified.

• Line 393: ... each of the filters. Corrected.

• Line 403: refer to the corresponding figures in this sentence and the previous sentence. Done.

• Figure D2: is 6300 opd in cm? Also could you add a legend describing colors? Also mention the date and time of measurements in the caption. Done

• Line 411: In resolution Fig D1 cm... not clear what you are referring to here! Corrected.

• Figure E1: this caption doesn't look like describing this figure. Corrected.

• Table F1: Describe the vibrational and rotational levels corresponding to each digit. Done.

• Figure G3: caption doesn't match the figure. Corrected.

• Figure H4: InGaAs spectra: which spectral window? Corrected.

• Figure H5: which one is on top and which one at the bottom? Corrected.

INSITU NOT AIRMASS

---

## Author Comment (AC2)

**Response to reviewer 2**

The authors appreciate the comments and corrections of the reviewers.

Specific comments:

The optical filter used for this study differ from the NDACC standard filter set. NDACC filter #1 covers the region from 3850–4550 cm-1 and does not include the 4800 cm-1 micro window. The filter used in this study is optional and in NDACC numbering (increasing number with increasing wavelength) something like NDACC filter #0. Therefore, it is questionable whether many sites recorded spectra with it since a long time.

Indeed is not a guarantee that this filter is being used in other sites. Ny-Alesund has one of the longest time series in the network and was the primary focus. Adding the MIR retrieval would add 7 additional years (1997-2004) of xCO2 data and also there is the potential for expanding geographically in future plans. E.g. Lauder has been measuring in this spectral range since mid 2020.

Since the MIR does not include O2 signatures pressure data have been used to calculate XCO2. Accordingly, the quality of the XCO2 product relies on the quality of the pressure sensor and its data. The paper would benefit from a paragraph or appendix describing the used pressure sensor and its requirements. In particular, the required precision and accuracy and any means to check and correct for drifts need to be discussed in the paper.

Great suggestion, a new appendix section will be added.

The paper includes an error estimate and the precision of the retrieved XCO2 product is good. Normally, a correlation coefficient of 0.95 (Fig. 11) is quite good, too. For XCO2 data however, it is not fully clear whether this is sufficient. For Burgos site, the correlation coefficient (0.84) is even lower.

Yes, the correlation is lower for Burgos, and we have two hypotheses as to why. Firstly, because the w4790 window has a higher temperature dependence, and secondly the position of the tropopause in the a priori profiles doesn't match the height of the real tropopause. These 2 factors combined result in a less accurate temperature profile leading to higher errors and lower accuracy of the retrieved xCO2. This accuracy is something that needs to be further improved.

It is certainly a good idea to include spectra from a wet site. The site used for this, however, suffers from the lack of NDACC spectra. Instead, spectra were recorded with a broadband InGaAs detector without any optical filter. In the 4800 cm-1 region an NDACC type spectrum recorded with a cooled InSb detector and bandwidth filter is better with respect to linearity and signal to noise ratio.

If this is the case then it will improve the accuracy of the retrieval in warm wet places. And it could be an additional explanation to the above discussed comment. Certainly it would be an interesting follow up study.

You might add the signal to noise ratio of the InSb and InGaAs spectra as shown in Figs. C1&C2. This might give a hint on the noise level although the spectral resolution differs strongly. Alternatively, did you record these spectra (C1&C2) with the same spectral

Resolution?

The signal to noise ratio has been added in the text . The spectra used for the smoothing is the same 3 years used in the rest of the paper, 2016 to 2018. So 6300 and w4790 InGaAs have the same resolution and they are retrieved from the same TCCON spectra. But the w4790 InSb has a different resolution (180 cm OPD) as it was measured with NDACC specifications.

The spectral resolution of 0.005 cm-1 as used in the NDACC mode is probably overdone in this spectral regime at 4800 cm-1. (At this spectral regime, my personnel guess would be something about 90 cm OPD or a resolution of about 0.01 cm-1, respectively.) The authors tested different spectral resolutions (Appendix D) which is a very useful exercise. However, I do not see a clear statement on the optimal resolution at 4800 cm-1. I would suggest adding a plot of a CO2 line recorded with different OPD to study the needed OPD to resolve the line fully.

I agree, this is why I performed the resolution tests. I added a plot with the suggested resolution. The final suggested resolution is the same as TCCON of 0.02 cm-1 because as you mention it is the resolution that allows the lines to be fully resolved.

In the introduction, the paper states that previously published recipes for XCO2 retrievals does not yield XCO2 data of sufficient quality to use the data for atmospheric research. This is correct. However, in the discussion or conclusion the results of the data set retrieved in this paper have not been compared to those previous data sets to demonstrate the improved data quality.

We have added a short comparison to the conclusion:

The retrieval proved to not have a big dependence on the a priori to correctly represent the daily and seasonal cycles (with correlation coefficients of at least 0.980 for all 3 retrievals:). **This is an improvement from Buschmann et al. (2016) where the averaging kernels limited the sensitivity making the retrieval highly dependent on the a priori. The retrieval also proved useful to retrieve daily and subdaily values, which is a different goal and purpose of Barthlott et al. (2015) study.** However…

Line 390

Furthermore, the long-term stability of a time series of this data product is not studied in detail. This should include a discussion of a possible drift of the pressure sensor. See comment above. Secondly, regular cell measurements to retrieve the ILS (Instrumental Line Shape) of the spectrometer are strongly recommended over the entire time series. Are ILS measurements made and used in this study and if so what are the ILS results? ILS parameters are also missing in the error estimate (Table 4).

The pressure sensor details and drift discussion has been added in Appendix F.

The ILS is not used in this study. The instrument is regularly aligned to meet TCCON requirements. This includes monthly cell measurements to monitor its performance. But indeed the ILS would affect the xCO2 retrieval, and that the magnitude of it should be characterised along with the ADCF when the full w4790 xCO2 product is produced.

While chapter 6 announces historical data from 1997 to 2018, Figs. 9, 10 and B1 just show data for the years 2016, 2017 and 2018. It seems this is limited to the availability of data from the 6300 cm-1 spectral region, isn't it? The NDACC type measurements at 4800 cm-1 started in 1997. When did the TCCON measurements at 6300 cm-1 started at Ny-Ålesund? It would be helpful for the reader if you add a table or columns to Table 1 listing the periods of available and used spectra for each retrieval and chapter. Also for Table 3 and Fig.7 the period covered would be helpful.

The historical data covering since 1997 is only in Figure 12.
The other plots in the time series, Figures 9 to 11, use the error weighted daily means for the periods 2016-2018 for Ny Alesund and 2017-2018 for Burgos as it is the overlapping years for the comparison.
In the rest of the paper's analysis, the xCO2 from InGaAs spectra (for both 6300 and w4790) use the same 3 and 2 years correspondingly.
But for InSb, only in section 4.2, I chose to use data from 1997 to 2018  to compensate for the lower quantity  of measurements in comparison to TCCON. I have clarified the years of data used for each section. Also included in Table 1 is the actual availability of data for each spectra type.
There is a larger availability of data for 6300 as described in section 2.1  but it was decided to use a limited range for the study.

Finally, if earlier TCCON data (from before 2016) are available it would be nice to include a longer data set into the comparison. Moreover, to study the long-term stability in more detail it would be good to calculate the difference of NDACC and TCCON XCO2 data from the beginning of these measurements and to perform a trend study on the difference.

Initially, we tried to keep the time series consistent between Burgos (available only since 2017) and Ny-Ålesund and are confident that the number of data points allows for meaningful comparison. We certainly see the benefit in performing the analysis not only on longer time scales but with additional sites. However due to time constraints we suggest this effort be taken in a follow up study.

Technical corrections:
- l. 20: validation validation
- l. 20: ofCO2
- l. 25: sppectru, => spectrum
- l. 59: In NDACC spectra => NDACC spectra
- l. 126: Correct use of respectively?
- l. 141 and a few more times there is still a question mark
- l. 158: Ny-Ålesund around => Ny-Ålesund and around
- l. 163: following => following equation
- l. 169: when => even if?
- l. 185: => (Wunch et al., 2011a)
- l. 226: the those => those?
- l. 229: These bias => These biases or this bias
- l. 311: is and full stop are missing.
- l. 326: points. . => points.

- l. 407: Fig. D1 is missing or the number of Fig. D2 is incorrect.
- l. 412: where => were or better rephrase the sentence beginning with 'In resolution Fig D1'
- l. 445: Ny-Ålesun => Ny-Ålesund
- l. 449: One authors => One author

All corrected.

The list of typos is not complete. Please proofread carefully.